# UNIPROMPT-CL: SUSTAINABLE CONTINUAL LEARNING IN MEDICAL AI WITH UNIFIED PROMPT POOLS

## ABSTRACT

Although modern AI models achieve state-of-the-art performance with large-scale datasets, strict ethical and institutional constraints in medicine make centralised learning nearly impossible. Institutions must therefore rely on local data, but traditional training methods quickly overfit new samples and suffer from catastrophic forgetting, making continual learning (CL) essential. While CL has advanced in the field of natural images, prompt-based continual learning (PCL) remains largely unexplored in the context of medical applications. We present UniPrompt-CL, the first PCL framework designed specifically for healthcare. Preliminary experiments show that existing PCL approaches perform poorly on medical datasets, which motivates our hypothesis that the prompt pool design needs to be more effective. UniPrompt-CL introduces a unified prompt pool with minimal expansion and a novel regularisation term, reducing computation while balancing stability and plasticity. On three diabetic retinopathy datasets (APTOS, DDR and DRD), UniPrompt-CL improves accuracy by at least 10% and the F1 score by 9 points compared to previous methods, while reducing the cost of inference. Additionally, it achieves superior performance on continual learning evaluation metrics. These results demonstrate that UniPrompt-CL lays the foundation for sustainable medical AI, enabling consistently high performance in distributed healthcare environments. To ensure reproducibility, the code and all training configurations can be found in this repository.

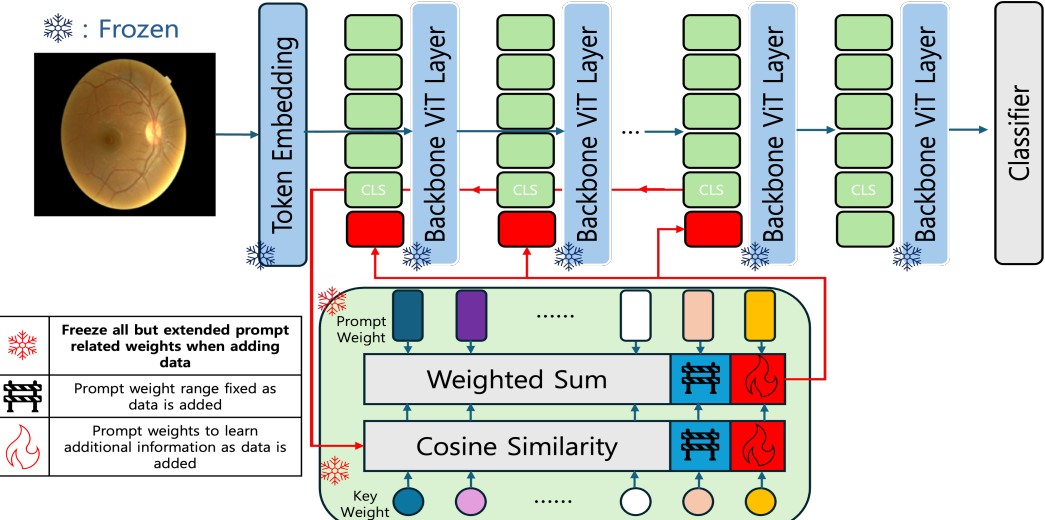

Figure 1: This figure presents the overall architecture proposed in this study, which integrates an enhanced prompt pool. At each layer, the [CLS] token serves as a query, and the resulting layer-wise queries are centrally managed through the prompt pool integration module. This integration generates the prompts for the subsequent layer, which are then combined with $x_l$ and propagated forward. Additionally, at each training stage, only a small number of new prompts are introduced through a minimal prompt expansion mechanism, while all previously learned prompts remain frozen.

## 1 INTRODUCTION

Modern artificial intelligence (AI) models achieve their best performance when trained on large-scale, high-quality datasets; however, in the medical domain, ethical, social, and institutional constraints make it exceedingly difficult to obtain such data Gerke et al. (2020); Herington et al. (2023). Variations in disease prevalence and restrictions on external data transfer render the collection of medical data and centralized training practically impossible Tschider et al. (2024). Furthermore, medical data is collected through various diagnostic devices at each hospital, and the prevalence rates collected also differ, frequently causing shifts in data distribution. This leads to degraded model performance in real-world settings. Indeed, a Google developed medical AI model demonstrated excellent performance during development but suffered abrupt drops in accuracy upon deployment in clinical settings Beede et al. (2020); Qazi et al. (2024). Consequently, there is a growing need for approaches that can overcome these practical challenges.

In this work, we focus on CL, which enables models to incrementally learn from data generated independently at each hospital, taking into account the unique characteristics of medical data. CL effectively incorporates new data while retaining previously acquired knowledge in settings subject to domain shifts and continuously evolving environments. Given the extreme restrictions on data sharing among hospitals, updating models locally at each institution is essential Ravishankar et al. (2019). However, traditional AI training methods, when unable to access prior data, are prone to catastrophic forgetting during sequential learning Abdelsalam et al. (2021); Agarwal et al. (2022); Belouadah & Popescu (2020); Bayasi et al. (2024); Kumari et al. (2023).

When applying CL in medical settings, several factors must be considered. First, medical imaging devices are heterogeneous rather than uniform, making it critical to explicitly track inter device differences Guan & Liu (2021); Kushol et al. (2023); Cleland (2023). Additionally, physiological variations among patients can introduce subtle color and texture differences in images that must be managed effectively. Although domain adaptation and generalization techniques can partially mitigate these issues, they remain insufficient to fully address catastrophic forgetting induced by continuous data shifts.

To address these challenges, we adopt a PCL framework that prioritizes both computational efficiency and data privacy. Existing PCL methods often require repeated Vision Transformer (ViT) inferences, or introduce additional query generation steps that impose considerable overhead Wang et al. (2022b); Smith et al. (2023); Menabue et al. (2024). Although recent efforts Kim et al. (2024b) have attempted to mitigate these limitations, they do not provide a comprehensive solution. Moreover, to the best of our knowledge, most existing PCL approaches have focused primarily on natural images. Therefore, to develop a PCL approach tailored to the medical domain, we build upon and extend the method proposed in Kim et al. (2024b). Specifically, we introduce a unified prompt pool that merges the layer-wise prompt sets into a single shared pool. During each training stage, only a minimal subset of prompts is expanded, while the remainder are fixed to minimize redundancy and alleviate catastrophic forgetting. Additionally, we propose a novel regularization term to promote effective and stable learning. Experimental results on three public diabetic retinopathy datasets Karthik et al. (2019); Li et al. (2019); Dugas et al. (2015) demonstrate that our method significantly outperforms state-of-the-art (SOTA) approaches, achieving higher accuracy and F1-scores while substantially reducing computational cost, and also delivering superior performance on continual learning metrics. The main contributions of this study are as follows:

• We consolidate layer wise prompt pools into a unified prompt pool and propose a new regularization term to train it effectively.
• By employing a single backbone instead of multiple models, we dramatically reduce computational cost while outperforming existing SOTA methods.
• We introduce a PCL strategy tailored to the medical domain that relies on expanding only a small number of prompts.

## 2 RELATED WORK

To position our work within the landscape of CL, we review recent surveys Qazi et al. (2024); Wang et al. (2024b) and categorize existing methods into four main classes. **Regularization Based Methods:** Regularization based approaches introduce penalty terms into the loss function to encourage

the model to retain important weights from previous tasks. Examples include Elastic Weight Consolidation (EWC) Kirkpatrick et al. (2017) and Synaptic Intelligence Zenke et al. (2017), which estimate the importance of each parameter and penalize changes proportionally Wang et al. (2024a); Pham et al. (2022). While effective on low dimensional or simple datasets, these methods struggle when data complexity and dimensionality increase, as the approximation of parameter importance becomes unreliable. **Architecture Based Methods:** Architecture based strategies dynamically adapt the network structure to accommodate new tasks. Techniques such as Progressive Neural Networks and Supermasks allocate new subnetworks or masks per task, offering strong scalability Ebrahimi et al. (2020); Wortsman et al. (2020). However, they incur growing memory and computation costs with each new task, making them less practical for long sequences of tasks or resource-constrained environments. **Rehearsal Based Methods:** Rehearsal based approaches store a small buffer of past examples and replay them alongside new data to prevent forgetting. Methods like iCaRL Rebuffi et al. (2017) and Dark Experience Replay Buzzega et al. (2020) have demonstrated strong performance on image benchmarks Wu et al. (2018); Shin et al. (2017). Yet, the need to store real data raises serious privacy and storage concerns particularly acute in the medical domain, where data sharing is heavily restricted. **PCL Methods:** PCL method fixes the majority of the pretrained model's parameters and learns only a small set of prompt vectors for each task. This design preserves existing knowledge while adapting efficiently to new domains, drastically reducing both memory and computation overhead  Chen et al. (2024); Wang et al. (2022b); Smith et al. (2023); Kim et al. (2024b). These methods have shown promise on natural-image benchmarks but often require multiple ViT inferences and have not yet been widely explored for medical data.

**The Need for Medical Domain PCL:** In summary, while regularization, architecture, and rehearsal methods each bring valuable insights, their applicability in the medical domain is limited by data complexity, resource constraints, and privacy issues. Prompt-based methods offer an appealing alternative by leveraging a fixed backbone and learning lightweight prompts. In contrast, our method enhances this paradigm by requiring only a single ViT inference and utilizing a unified prompt pool, rather than maintaining multiple separate ones.

## 3  PRELIMINARY

The primary objective of this paper is to efficiently mitigate domain shift arising from multiple hospitals and various sources, and to comprehensively address the drawback of existing PCL methods that require multiple backbones during training and inference. As noted earlier, we adopt a PCL approach to solve these issues. Our proposed methodology builds upon the recently introduced One-Stage PCL framework Kim et al. (2024b), extending it with key improvements. While all unspecified settings follow those in the referenced work, **our approach differs in two main aspects**: (1) it achieves high performance with only a single ViT inference, and (2) it introduces a unified prompt-pool design and training strategy tailored to medical data, inspired by insights from our experiments.

**Continual Learning Problem Setting :** In this section, we briefly describe the CL environment. We consider a scenario in which data collected at each hospital are learned via CL. To this end, we utilize publicly available datasets collected from different hospitals with identical disease and severity labels  Karthik et al. (2019); Li et al. (2019); Dugas et al. (2015). During training, our CL environment sequentially and independently learns from data with dynamic distributions at each stage, meaning that samples from previous stages or future data are inaccessible. We also assume no rehearsal buffer is used, following the rehearsal-free CL paradigm defined in  Wang et al. (2024b). Furthermore, our experiments are conducted under a domain-incremental learning (DIL) setting where task identities are unknown.

**Prompt-based Continual Learning (PCL) :** To understand this paper, it is important to examine the foundational OS-Prompt framework and its extended version, OS-Prompt++ Kim et al. (2024b). Both frameworks aim to overcome the limitations of existing PCL methods. Traditional PCL approaches often incur high computational costs due to the use of an additional query function (e.g., a ViT) to generate prompt queries. To address this, OS-Prompt was proposed as a lightweight alternative that eliminates the query function, thereby significantly reducing computational overhead. The process operates as follows:

$$q_l = x_{l[\text{CLS}]}, \quad q_l \in \mathbb{R}^{1 \times D} \tag{1}$$

Where, $x_l \in \mathbb{R}^{N \times D}$ denotes the input token embeddings at layer $l$, $N$ is the number of input tokens and $D$ is the embedding dimension. To select prompts from the existing prompt pool, a cosine similarity based weighted sum is used. The prompt key matrix $K_l$ is formed by stacking the $L_p$ prompt key vectors $\left[k_l^1, k_l^2, \ldots, k_l^{L_p}\right]$ and transposing, yielding a matrix in $\mathbb{R}^{L_p \times D}$. Where, $L_p$ is the total number of prompts, $k_l^m$ is the key vector of the $m$-th prompt ($k_l^m \in \mathbb{R}^{1 \times D}$), and the corresponding prompt value vector is $p_l^m \in \mathbb{R}^{1 \times D}$. Using the cosine similarity function $\gamma(\cdot)$, the selected prompt is computed as follows.

$$\hat{\phi}_l = \sum_{m=1}^{L_p} \gamma(q_l, k_l^m)\, p_l^m, \quad \hat{\phi}_l \in \mathbb{R}^{1 \times D} \tag{2}$$

However, OS-Prompt relies solely on a single [CLS] token from an intermediate layer, which limits its representational capacity and leads to only modest performance gains. To address these limitations, OS-Prompt++ requires an additional step during training: reintroducing query functionality to enhance the expressiveness of the [CLS] token.

In other words, OS-Prompt++ still requires two ViT inferences during the learning process, making it difficult to consider it a true single-backbone solution.

## 4 PROPOSED METHOD

Our primary objective is to achieve superior CL performance with a single backbone. To this end, our methodology is built upon three key components: (1) a unified prompt pool, (2) a minimal prompt expansion strategy, and (3) a consistency-enforcing regularization term.

As shown in Table 7 of Appendix A, medical datasets often exhibit severe long-tailed class distributions, which exacerbate catastrophic forgetting as training stages progress. To address this issue, we adopted the powerful DINOv2 backbone Oquab et al. (2023) and selected **DINOv2-base (86.6M parameters)** for its balanced trade-off between representational capacity and computational efficiency. For further details, please refer to Table 9 in Appendix D.

Our experiments demonstrate that this strong backbone, combined with our prompt-based design and regularization, effectively mitigates both data imbalance and catastrophic forgetting in medical CL scenarios. The overall architecture of our approach is illustrated in Figure 1.

### 4.1 MOTIVATION FOR PROPOSAL AND INTEGRATION OF PROMPT-POOL

**Motivation:** Prior studies Guan & Liu (2021); Kushol et al. (2023); Cleland (2023) show that medical images are acquired under standardized protocols and that tracking subtle lesions is essential for diagnosis. From this, we hypothesize that prompt learning in medical imaging requires fine-grained adjustments. To validate this, we visualized prompts from existing PCL methods. As shown in Figure 2.a, the brown dots represent prompts in our unified pool, while other colors correspond to OS-Prompt. The visualization shows that existing PCL prompts are widely dispersed, making it difficult to capture subtle details needed in medical images. Appendix C provides further examples and clear evidence of why PCL fails on medical data. Building on this, we propose a unified prompt pool that allows all layers to share prompts, maximizing fine-grained tracking across the hierarchy.

**Method:** Having established the motivation, we now describe the structure of our Integration of Prompt-Pool. Following Equation (1), we use the [CLS] token embedding of each layer as the prompt query $q_l$. The prompt keys are unified and defined as $k^m \in \mathbb{R}^{1 \times D}$, and the prompt value vectors are unified and defined as $p^m \in \mathbb{R}^{1 \times D}$. Consequently, the prompt computation is reformulated as follows:

$$\phi_l = \sum_{m=1}^{L_p} \gamma(q_l, k^m)\, p^m, \quad \phi_l \in \mathbb{R}^{1 \times D}. \tag{3}$$

The resulting prompt $\phi_l$ is then unified with the layer transformation:

$$x_{l+1} = f_l(x_l, \phi_l). \tag{4}$$

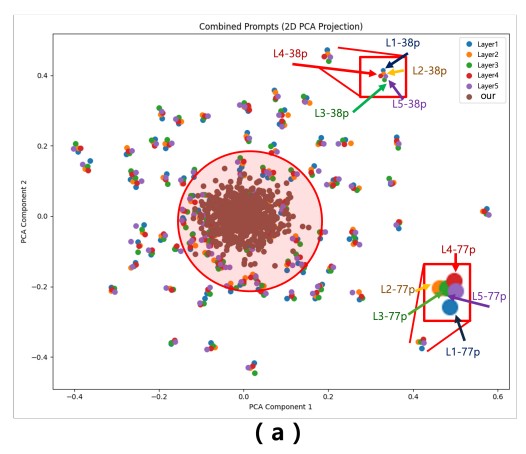 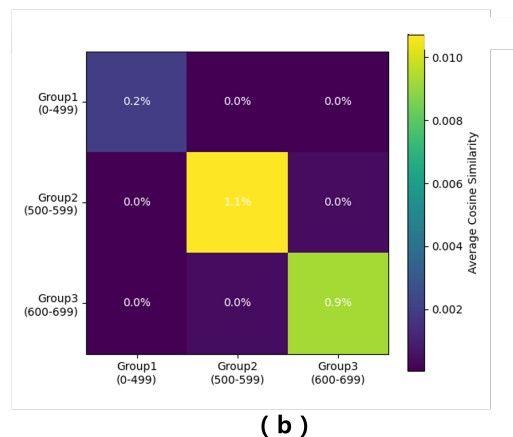

( a )  ( b )

Figure 2: (a) visualizes the prompts of OS-Prompt (independent pool) and our proposed approach (integrated pool). The dots corresponding to layers 1 through 5 represent the prompts from OS-Prompt, while the brown dots (Ours) indicate the prompts generated by our integrated pool. Refer to the upper-right corner of the figure for further details. (b) shows that, as training stages progress, the newly added prompts do not learn redundant or overlapping information, suggesting that each prompt captures distinct features.

## 4.2 FEW PROMPT EXPANSION

In this subsection, we introduce the *Few Prompt Expansion* strategy. This strategy serves as an alternative to multi-ViT inference, maximizing the use of existing prompt information while enabling the acquisition of additional knowledge required for domain expansion. When transitioning to the next CL stage (dataset), we first construct and train the integrated prompt pool as described in Section 4.1, and then freeze the prompt weights corresponding to the previous stages. Next, the prompt set is expanded by adding 20% of the original $L_p$ prompts. This ratio was empirically determined in Table 10 of Appendix E. The newly added prompts are denoted as $\psi$, resulting in a total of $L_p + \psi$ prompts at each stage. The combined set of prompts is represented as $\hat{L}_p$, whose size varies depending on the stage. However, the newly added prompts ($\psi$) may initially have less influence compared to well-trained prompts, which can limit their learning effectiveness. To address this issue, we introduce the following formulation to ensure that the added prompts more effectively capture new information.

$$z_l = \text{Softmax}\left(\frac{K\, p_l^\top}{\|K\|_2\, \|p_l\|_2}\right), \quad z_l \in \mathbb{R}^{B \times \hat{L}_p}, \tag{5}$$

We then refine this to:

$$S_{b,m} = 1 - z_{b,m}, \quad \mathcal{L}_s = \frac{1}{B \times \hat{L}_p} \sum_{b=1}^{B} \sum_{m=1}^{\hat{L}_p} S_{b,m}, \tag{6}$$

where $B$ is the batch size and $\hat{L}_p$ the total number of prompts.

Finally, the overall loss function is defined as:

$$\mathcal{L}_{\text{total}} = \mathcal{L}_{\text{CE}} + \lambda\, \mathcal{L}_s, \tag{7}$$

where $\lambda$ is a hyperparameter set to 0.001 in this study to control the strength of the regularization term. The value of $\lambda$ is further discussed in detail in the ablation study.

## 5 EXPERIMENT SETTINGS

In this section, we describe our experimental setup in detail to substantiate our claims. First, we introduce the datasets used in our experiments; second, we outline the data augmentation and pre-

processing methods, baseline validation procedures, evaluation metrics, and implementation environment. Detailed information about the datasets and the experimental settings can be found in Appendix A.

**Dataset:** We simulate a DIL scenario using three publicly available datasets collected from different hospitals with identical disease and severity labels: APTOS 2019 (APTOS) Karthik et al. (2019), LI2019 (DDR) Li et al. (2019), and Diabetic Retinopathy Detection (DRD) Dugas et al. (2015). Prior to training, and following the procedure in Kobat et al. (2022), each dataset's original classes are consolidated into three joint labels. The datasets are then presented to the model in the sequence:

$$APTOS \rightarrow DDR \rightarrow DRD.$$

**Experimental Details:** During training, data augmentations including HorizontalFlip, Rotation, and VerticalFlip are applied at random. ViT preprocessing is performed using the AutoImageProcessor (weights: facebook/dinov2-base) from the transformers library; see Appendix Table 8 and Appendix Table 11 for detailed settings. Baseline methods are validated using the rigorously implemented CL library from Boschini et al. (2022); Buzzega et al. (2020), with any additional required code custom implemented. Standard CL approaches are often evaluated on large-scale datasets such as ImageNet or CIFAR-100. Consequently, prior studies typically quantify catastrophic forgetting using the standard CL performance matrix, including Backward Transfer (BWT), Final Average Accuracy (FAA), Average Accuracy (AvgACC), and Average Forgetting (AvgF). In our work, because we operate on limited-size and imbalanced medical datasets, we follow this protocol by reporting AvgACC, BWT, and AvgF after the final stage, and additionally report per-stage accuracy (Acc) and F1-score (F1) at every CL stage to more faithfully capture how performance evolves under class imbalance. The per-stage training procedure is instantiated once for each dataset in the continual-learning sequence; that is, the total number of stages equals the number of datasets. The reported results are averaged over three independent runs. For a concise, step-by-step pseudocode that makes the model workflow and CL staging explicit, see Algorithm 1 in the Appendix.

## 6 EXPERIMENT RESULTS

In this session, to aid the readers' understanding, we separate the results of each table and figure into individual sections and discuss them in detail.

**Stage-Wise Prompt Independence and Fine-Grained Feature Encoding :** As discussed in Section 4.2. Motivation, we hypothesize that the prompts added at each stage should not only retain previously acquired information but also capture new, fine-grained features. Figure 2.(b) supports this hypothesis by presenting a similarity matrix of the prompts introduced at successive stages under our proposed method. The clear diagonal pattern indicates that prompts are learned independently at each stage, with minimal redundancy across stages. In other words, newly added prompts effectively encode novel information without overlapping with existing ones. Additionally, Figure 2.(a) demonstrates that the expanded prompts focus on subtle distinctions. In this figure, the brown markers represent prompts learned through our proposed prompt pool, while markers of other colors represent prompts learned using the existing PCL method. Whereas the prompts from OS-Prompt broadly span the feature space, our integrated prompts form tighter clusters that reflect more refined feature representations. Finally, the quantitative results in Table 1 indicate that our method outperforms existing SOTA models, providing empirical evidence for the hypothesis presented in Section 4.1.

**Comparative Performance Analysis with SOTA CL Methods:** Table 1 contrasts our method not only with strong PCL baselines but also with representative non-PCL families (architecture-modified, rehearsal, and regularization), reporting final-stage accuracy and F1-score on each dataset. Overall, our method achieves higher accuracy and F1-score than the competing approaches, demonstrating superior performance in continual learning for the medical domain.

This improvement can be interpreted as being closely related to the prompt distribution shown in Figure 2.a. Existing SOTA PCL methods maintain a broad prompt region primarily because they were originally designed for natural image processing. In natural images, a wide variety of viewpoints, objects, and compositional changes are present, and capturing such comprehensive information leads to better performance. The example image can be found in Appendix Figure 3. Therefore, a broad prompt region is required to represent diverse features effectively.

Table 1: Final Accuracy (Acc) and F1-Score (F1) Results After the Final Stage and Performance Comparison with Other PCL Models. The symbol † denotes our proposed model. PCL denotes prompt-based continual learning; Arch-CL refers to architecture-based continual learning; Reh-CL means rehearsal-based continual learning; and Reg-CL indicates regularization-based continual learning. (**bold** indicates the highest performance; scores are mean values with negligible deviations.)

| CL-Type (Ref) | Model | APTOS | | DDR | | DRD | |
|---|---|---|---|---|---|---|---|
| | | Acc | F1 | Acc | F1 | Acc | F1 |
| PCL (ECCV2024) | OS Kim et al. (2024b) | 0.687 | 0.637 | 0.693 | 0.648 | 0.619 | 0.568 |
| PCL (ECCV2024) | OS++ Kim et al. (2024b) | 0.743 | 0.686 | 0.697 | 0.655 | 0.623 | 0.565 |
| Arch-CL (CVPR2024) | MoE-Adapters Yu et al. (2024) | 0.835 | 0.742 | 0.747 | 0.694 | 0.564 | 0.478 |
| PCL (CVPR2023) | Coda-Prompt Smith et al. (2023) | 0.682 | 0.646 | 0.721 | 0.697 | 0.663 | 0.557 |
| PCL (CVPR2022) | L2P Wang et al. (2022b) | 0.353 | 0.174 | 0.421 | 0.194 | 0.603 | 0.252 |
| PCL (ECCV2022) | Dual-prompt Wang et al. (2022a) | 0.363 | 0.185 | 0.435 | 0.222 | 0.604 | 0.259 |
| Reh-CL (NIPS2020) | DER++ Buzzega et al. (2020) | 0.531 | 0.442 | 0.609 | 0.567 | 0.681 | 0.612 |
| Reg-CL (ICML2018) | Online EWC Schwarz et al. (2018) | 0.746 | 0.695 | 0.702 | 0.708 | 0.698 | 0.653 |
| - | **UniPrompt-CL**† | **0.849** | **0.761** | **0.772** | **0.723** | **0.701** | **0.656** |

Table 2: Tracking and comparing various catastrophic forgetting outcomes during stage progression (where Red indicates data learned at the current step, Blue indicates previously learned data (seen), and Black indicates unseen data) [Accuracy (Acc), F1-score (F1), OS prompt++ (OS++); **Horizontal: Training Data, Vertical: Evaluation Data**]. Additionally, the FLOPs row indicates the amount of computing resources used. The symbol † denotes our proposed model.

| | | Evaluation | | | | | | | | | | | |
|---|---|---|---|---|---|---|---|---|---|---|---|---|---|
| | | OS-Prompt++ (Dual inference) | | | | | | UniPrompt-CL† (Single inference) | | | | | |
| Training | Dataset | APTOS | | DDR | | DRD | | APTOS | | DDR | | DRD | |
| | | Acc | F1 | Acc | F1 | Acc | F1 | Acc | F1 | Acc | F1 | Acc | F1 |
| Stage 1 | APTOS | 0.868 | 0.753 | 0.565 | 0.474 | 0.409 | 0.354 | 0.901 | 0.767 | 0.601 | 0.447 | 0.453 | 0.381 |
| Stage 2 | DDR | 0.707 | 0.638 | 0.797 | 0.748 | 0.508 | 0.413 | 0.866 | 0.663 | 0.878 | 0.844 | 0.636 | 0.534 |
| Stage 3 | DRD | 0.743 | 0.686 | 0.697 | 0.655 | 0.623 | 0.565 | 0.849 | 0.761 | 0.772 | 0.723 | 0.701 | 0.656 |
| **FLOPs** | | 66.42 GFLOPs | | | | | | 44.17 GFLOPs | | | | | |

In contrast, fundus images are acquired under strictly defined protocols with consistent framing and conditions, making it essential to capture subtle variations in color profiles arising from device and patient-specific characteristics. You can see image samples of natural images and medical images in Appendix Figure 3. As a result, while prompt clusters in natural images tend to be widely distributed, in medical imaging, a more fine-grained prompt distribution proves to be more effective. Our findings highlight that such domain-specific characteristics explain the performance limitations of existing SOTA PCL methods and underscore the necessity of tailored learning strategies for each domain.

**Stage-Wise Forgetting Mitigation and Performance Gains:** In this section, we describe the performance comparison results presented in Table 2. Table 2 builds upon the results of Table 1 and illustrates how accuracy and F1-score change across learning stages, with OS-Prompt++, the best-performing prompt-based continual learning method in Table 1, serving as the baseline. The most notable observation in Table 2 is that the proposed method not only mitigates catastrophic forgetting effectively but also improves overall performance. Moreover, while OS-Prompt++ relies on two ViT inferences, our approach achieves superior performance with only a single ViT inference. Furthermore, even when employing expanded prompts, our method requires only one ViT inference, thereby significantly reducing FLOPs and demonstrating high computational efficiency.

## 7 ABLATION STUDY

In this section, we conduct ablation studies to improve interpretability and reliability by clarifying the causal factors behind performance gains. Our objective is to pinpoint directions for model improvement and explicitly highlight the key contributing components. In addition, we perform

small-scale domain-incremental experiments on three skin-cancer datasets to assess generalization beyond DR, providing evidence that the proposed components transfer reliably across different medical imaging domains.

Table 3: We compare the results of fatal forgetting during the stepwise progression of the baseline model, the introduction of a stronger backbone, and the methodology of this study. Through these exclusion studies, we highlight the importance of a good backbone in PCL and show that there is room for further improvement. It can be interpreted in the same way as Table 2.

**OS-Prompt++ (Original)**

|  | APTOS | | DDR | | DRD | |
|---|---|---|---|---|---|---|
|  | Acc | F1 | Acc | F1 | Acc | F1 |
| APTOS | 0.868 | 0.753 | 0.565 | 0.474 | 0.409 | 0.354 |
| DDR | 0.707 | 0.638 | 0.797 | 0.748 | 0.508 | 0.413 |
| DRD | 0.743 | 0.686 | 0.697 | 0.655 | 0.623 | 0.565 |

**OS-Prompt++ (Add Dino-v2)**

|  | APTOS | | DDR | | DRD | |
|---|---|---|---|---|---|---|
|  | Acc | F1 | Acc | F1 | Acc | F1 |
| APTOS | 0.918 | 0.823 | 0.608 | 0.520 | 0.492 | 0.467 |
| DDR | 0.732 | 0.604 | 0.849 | 0.828 | 0.625 | 0.563 |
| DRD | 0.754 | 0.690 | 0.763 | 0.721 | 0.668 | 0.585 |

**UniPrompt-CL[†] (Proposed)**

|  | APTOS | | DDR | | DRD | |
|---|---|---|---|---|---|---|
|  | Acc | F1 | Acc | F1 | Acc | F1 |
| APTOS | 0.901 | 0.767 | 0.601 | 0.447 | 0.453 | 0.381 |
| DDR | 0.866 | 0.663 | 0.878 | 0.844 | 0.636 | 0.534 |
| DRD | 0.849 | 0.761 | 0.772 | 0.723 | 0.701 | 0.656 |

**Effectiveness of Strong Backbones in Prompt-Based Continual Learning:** In Table 3, we demonstrate both the necessity of incorporating a powerful backbone into PCL and that our method's superior performance cannot be attributed to the backbone alone. We examine foundation models such as Dino-V2 because, unlike conventional backbones, these models are pretrained on massive datasets and are capable of extracting high-quality features across diverse tasks. Although several prior studies have utilized foundation models to achieve outstanding few-shot and zero-shot performance Kirillov et al. (2023); Alayrac et al. (2022); Kim et al. (2024a); Singh et al. (2025); Ren et al. (2024), their applicability within the PCL domain remains underexplored. Given the severe class imbalance and limited sample sizes commonly observed in the medical domain, we anticipated greater performance gains from adopting a foundation model. Accordingly, to test our hypothesis fairly within the PCL paradigm, we use OS-Prompt++ the top-performing PCL method in Table 1 as our baseline for comparison.

- Original OS-Prompt++ (without backbone enhancement)
- OS-Prompt++ with a Dino-V2 backbone
- Our proposed method

To ensure a fair comparison, all experimental conditions were held constant except for the backbone. Comparing the original OS-Prompt++ with its Dino-V2 variant, we observe that the incorporation of Dino-V2 improves both overall accuracy and robustness against forgetting. This result indicates that a stronger backbone provides generalization benefits analogous to few-shot and zero-shot learning, even within PCL settings. Importantly, our full method still outperforms both variants, demonstrating that the gains achieved by Few Prompt Expansion and Integration of the Prompt Pool extend beyond what can be attributed to backbone strength alone. Table 4 presents additional experiments and metrics, and the evaluation of CL metrics is elaborated there and in the following analysis.

Table 4: Performance evaluation of AvgACC, BWT, and Cost-Adjusted Retained Accuracy (CARA) across three diabetic retinopathy datasets. Training-time GFLOPs. (per step, including forward and backward passes) The symbol † denotes our proposed model. The best results are highlighted in **bold**.

| Method | AvgACC↑ | BWT↑ | AvgF↓ | GFLOPs↓ | $CARA_{0.5}$ ↑ |
|---|---|---|---|---|---|
| OS-Prompt | 0.666 | -0.132 | 0.132 | **34.26** | 0.098 |
| MoE-Adapters | 0.716 | -0.080 | 0.080 | 105.64 | 0.064 |
| Coda-Prompt | 0.688 | -0.140 | 0.140 | 134.33 | 0.051 |
| L2P | 0.459 | -0.296 | 0.296 | 116.93 | 0.029 |
| Dual-prompt | 0.467 | -0.291 | 0.291 | 105.05 | 0.032 |
| DER++ | 0.607 | -0.288 | 0.288 | 168.02 | 0.033 |
| Online EWC | 0.715 | -0.174 | 0.174 | 100.62 | 0.059 |
| OS-Prompt++ (Original) | 0.769 | -0.113 | 0.113 | 51.12 | 0.095 |
| OS-Prompt++ (Add Dino-v2) | 0.812 | -0.125 | 0.125 | 66.42 | 0.087 |
| **UniPrompt-CL†** | **0.844** | **-0.079** | **0.079** | 44.17 | **0.116** |

**Quantitative validation using forgetting metrics:** In this subsection, we evaluate all models appearing in Tables 1 and 3 using both standard CL metrics and the Cost-Adjusted Retained Accuracy (CARA). A higher CARA indicates more stable and efficient performance in standard CL per unit GFLOPs. The definitions and computation procedures for the standard metrics and CARA are summarized in Appendix B. As shown in Table 4, our method generally outperforms PCL, regularization, rehearsal, and architecture-based baselines. Notably, it consistently surpasses the backbone-strengthened PCL, suggesting that the gains stem from the fixed-ratio few-prompt expansion and the unified prompt-pool design. Finally, from the CARA perspective, our approach exceeds all SOTA methods, demonstrating the best performance to GFLOPs efficiency. The best results in Table 4 are highlighted in **bold**.

**Impact of Prompt Expansion Ratio on Continual Learning Performance:** In this subsection, we discuss the procedure for selecting the prompt expansion ratio. We evaluated candidate ratios of 10%, 20%, and 30%, and the results are presented in Appendix Table 10. All experiments adhered to the detailed hyperparameter settings outlined in Appendix Table 8. We ultimately selected a 20% expansion ratio because increasing from 10% to 20% yielded clear performance improvements, whereas further expanding to 30% increased the parameter count without corresponding gains and even caused slight performance degradation. These findings indicate that indiscriminate expansion of parameters does not guarantee better performance.

**Impact of the Proposed Loss Term:** In this subsection, we evaluate the performance impact of the loss term proposed in Section 4.2. The results are presented in Table 5, where all metrics are measured after the final training step. Where, FAA denotes the Final Average Accuracy across all stages, and FAF indicates the Final Average F1-score across all stages. The hyperparameter configurations follow the settings listed in Appendix Table 8. First, the results obtained without the proposed loss term consistently show performance degradation compared to those with the loss term applied. This confirms that the proposed loss operates as intended, demonstrating robustness and efficiency, while also suggesting potential for further improvement. Furthermore, we analyze the effect of the $\lambda$ value used in the loss formulation. Interestingly, larger values of $\lambda$ tend to increase FFA, whereas smaller values of $\lambda$ tend to increase FAF. Since both metrics are equally important, we finally adopt $\lambda = 0.001$ as a balanced choice.

Table 5: We compare the performance with and without $\mathcal{L}_s$ under various values of $\lambda$ (Eq. 7). The FAA and FAF after the last stage are reported.

| $\mathcal{L}_s$ | $\lambda$ | FAA | FAF |
|---|---|---|---|
| ✗ | ✗ | 0.754 | 0.701 |
| ✓ | 0.01 | **0.777** | 0.705 |
| ✓ | 0.001 | 0.775 | 0.713 |
| ✓ | 0.0001 | 0.765 | **0.723** |

Table 6: Performance evaluation of AvgACC and BWT across three small external skin canser datasets. The symbol † denotes our proposed model.

| Model | AvgACC↑ | BWT↑ | AvgF↓ | GFLOPs↓ | $CARA_{0.5}$ ↑ |
|---|---|---|---|---|---|
| OS | 0.682 | -0.135 | 0.135 | **34.26** | 0.101 |
| OS++ | 0.725 | -0.063 | 0.063 | 51.12 | 0.095 |
| MoE-Adapters | 0.597 | -0.040 | 0.040 | 105.64 | 0.056 |
| Coda-Prompt | 0.713 | -0.041 | 0.041 | 134.33 | 0.059 |
| Dual-prompt | 0.637 | **-0.012** | **0.012** | 105.05 | 0.061 |
| DER++ | 0.722 | -0.099 | 0.099 | 168.02 | 0.050 |
| Online EWC | 0.708 | -0.157 | 0.157 | 100.62 | 0.060 |
| UniPrompt-CL† | **0.732** | -0.049 | 0.049 | 44.17 | **0.105** |

**External Validation on Additional Datasets:** To further verify that our framework is not restricted to diabetic retinopathy (DR) but is generally applicable to medical imaging tasks, we additionally conduct a small-scale pilot study on skin cancer classification. Specifically, we construct a continual learning scenario using three dermatology datasets—ISIC Rotemberg et al. (2021), HAM Codella et al. (2019), and DERM7 Kawahara et al. (2018)—and provide dataset statistics and full experimental protocol in Appendix H.

In this setting, our method achieves the highest AvgACC on the skin cancer benchmarks, indicating that the proposed framework can transfer beyond DR and remains effective on a distinct disease and imaging domain. While our BWT/AvgF are sometimes slightly below a few baselines—indicating mild forgetting the overall utility is decisively better once compute is accounted for: Our method attains the highest AvgACC (0.844) at single-pass 44.17 GFLOPs, and its cost-adjusted retained accuracy $CARA_{0.5} = 0.105$ surpasses strong PCL baselines—OS (0.101), OS++ (0.095), and Coda-Prompt (0.059)—and also exceeds representative non-PCL families, including rehearsal-based DER++ (0.05), regularization-based Online EWC (0.06), and architecture-based MoE-Adapters (0.056). A complete summary of all models (AvgACC, BWT, AvgF, GFLOPs, and $CARA_{0.5}$) is provided in Table 6. In short, on a per-compute basis our method retains and delivers more accuracy, making the modest gaps in forgetting metrics a reasonable, principled trade-off rather than a substantive limitation.

## 8 CONCLUSION AND FUTURE WORK

In this work, we proposed UniPrompt-CL, a framework that combines a strong foundation backbone with efficient prompt management to mitigate catastrophic forgetting in medical AI. UniPrompt-CL outperforms existing SOTA CL methods, achieving over 10% accuracy improvement and a 9-point increase in F1 score across three diabetic retinopathy datasets. It also surpasses SOTA PCL techniques in the CL performance matrix. Notably, unlike prior PCL approaches that required multiple passes or complex pipelines, our framework exceeds their performance with only a single ViT inference. Our findings further highlight the importance of domain-specific considerations: unlike natural images, medical data follow standardized acquisition protocols, which substantially affect the behavior of PCL strategies. Existing methods generalized poorly to such data, whereas our design with an integrated prompt pool and lightweight expansion preserved prior knowledge and captured fine-grained features, yielding robust and stable performance.

While our large-scale evaluation focused on diabetic retinopathy classification, we additionally conducted a small-scale pilot study on skin cancer datasets, where UniPrompt-CL again achieved the strongest AvgACC, suggesting promising transferability to other medical imaging domains. Extending this line of investigation to further modalities (e.g., CT, MRI, pathology) and tasks such as segmentation or detection would provide stronger validation. Moreover, although Dino-V2 served as a strong backbone in our experiments, domain-specific vision encoders such as Med-CLIP or Med-ViT may further improve efficiency and accuracy in medical applications.

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

# APPENDIX

## APPENDIX OVERVIEW

This appendix provides supplementary details on implementation, evaluation metrics, medical data characteristics, model analysis, prompt expansion experiments, and the disclosure of generative AI usage.

- **A. Implementation and Reproducibility Details** Summary of experimental settings and reproducibility resources

- **B. Continual Learning Metrics and CARA**
  Standard CL metrics (AvgACC, FAA, BWT, AvgF) and the proposed compute-adjusted metric CARA.

- **C. Why Medical Data Requires Different Prompt Learning** Rationale for Unified Prompt Learning Strategies in Medical Continual Learning

- **D. Model Size Analysis: Facebook/DINOv2** Comparison of model variants and justification for backbone selection

- **E. Expansion Ratio We Selected** Experimental results on prompt expansion ratios and chosen configuration

- **F. UniPrompt-CL Training Procedure** Stage-wise training algorithm (pseudocode), including prompt freezing and fixed per-stage prompt expansion.

- **G. KEY HYPERPARAMETERS FOR BASELINES** Key hyperparameters for prompt-based continual learning baselines

- **H. Datasets for Small-Scale External Experiments** 3 skin-cancer datasets, domain-incremental; undersampled; same setup; 8:1:1 splits.

- **I. GenAI Usage Disclosure** Statement on the use of generative AI tools in manuscript preparation

## A  IMPLEMENTATION AND REPRODUCIBILITY DETAILS

Table 7: Distribution of datasets used for training

| Dataset | Normal (0) | Mild (1) | Severe (2) |
|---|---|---|---|
| APTOS Karthik et al. (2019) | 1805 | 1562 | 295 |
| DDR Li et al. (2019) | 6266 | 5343 | 913 |
| DRD Dugas et al. (2015) | 5000 | 8608 | 708 |

Table 8: Experiment Settings

| Item | Value |
|---|---|
| GPU | $2 \times$ V100 |
| RAM | 256GB |
| Learning Rate | $z0.001$ |
| Scheduler | Cosine Scheduler |
| Number of Prompts ($L_p$) | 500 |
| Prompt Dimension ($D$) | 768 |
| Optimizer | AdamW |
| Early Stop Patience | 5 |
| Epochs | 100 |
| Batch Size | 64 |
| Image Size | $224 \times 224$ |

To ensure the reproducibility and completeness of our experimental results, we provide all necessary resources and detailed information as follows.

**First**, we describe the datasets used for training. Our study leverages three publicly available diabetic retinopathy detection datasets, all sourced from Kaggle. Notably, some benchmarks do not provide ground-truth labels for their test sets. Following prior work Kobat et al. (2022), we applied preprocessing and re-partitioning strategies to create appropriate training, validation, and testing splits. In line with the clinician-authored study Kobat et al. (2022), we consistently remapped the original five classes—normal, mild DR, moderate DR, severe DR, proliferative DR—into three: normal, non-proliferative DR (NPDR), and proliferative DR (PDR), by merging mild/moderate/severe DR into a single NPDR class while keeping normal and PDR unchanged. The final data distribution and the exact sources of each dataset are summarized in Appendix Table 7, ensuring that our results remain directly comparable to related studies while staying faithful to the original data. The training data are denoted as

$$D = \{D_1, D_2, \ldots, D_n\}, \quad D_n = \{X_{n,b}, Y_{n,b}\}.$$

Where, $n$ indicates the stage index of the arriving dataset, $X_{n,b}$ represents an input image, and $Y_{n,b}$ denotes the corresponding ground-truth label in the image recognition task, where $b$ denotes the index within a batch of size $B$.

**Second**, we provide detailed information regarding the hardware configuration and hyperparameter settings used to implement and evaluate all experiments. Our implementation was developed using Python 3.8 with PyTorch **2.4.1+cu118**. All experiments were conducted in the controlled environment described in Appendix Table 8. For training efficiency, we employed two NVIDIA V100 GPUs; however, it is worth noting that the full GPU memory capacity was not utilized. This is because PCL methods refrain from retraining the majority of the backbone, updating only the additional learnable parameters. In practice, training on a single V100 GPU required approximately 8GB of VRAM, confirming the lightweight nature of our approach and the feasibility of reproducing results even on modest hardware.

**Finally**, In line with our commitment to reproducibility, all dataset partition files, preprocessing scripts, and training code can be found at this repository. This will enable the community to fully replicate our results, verify our findings, and further extend UniPrompt-CL for broader medical continual learning applications.

## B    CONTINUAL LEARNING METRICS

$$\text{AvgACC} = \frac{1}{T} \sum_{i=1}^{T} \left( \frac{1}{i} \sum_{j=1}^{i} R_{i,j} \right). \tag{8}$$

**Symbols:** - $T$: total number of stages (or tasks) - $R_{i,j}$: accuracy on task $j$ after training up to stage $i$

**Explanation:** This metric measures the average accuracy trajectory throughout the entire training process. After learning each task $i$, it computes the average accuracy over all tasks learned so far (from 1 to $i$), and then averages this value across all $T$ tasks.

**Meaning:** AvgACC provides a comprehensive view of how stably the model maintains its performance over time. Unlike metrics that only evaluate the final result, it reflects performance consistency across the whole learning process.

$$\text{FAA} = \frac{1}{T} \sum_{j=1}^{T} R_{T,j}. \tag{9}$$

**Symbols:** - $T$: total number of stages - $R_{T,j}$: final accuracy on task $j$ after completing all $T$ stages

**Explanation:** FAA measures the average accuracy across all tasks at the final training stage $T$.

**Meaning:** FAA directly shows how well the model retains knowledge of previously learned tasks after completing all training, making it a primary indicator of resistance to catastrophic forgetting. In addition, due to the severe class imbalance inherent in medical data, we also computed the above

metrics using the F1-score, which we refer to as FAF (Final Average F1-score), and employed it as an auxiliary evaluation metric.

$$\text{BWT} = \frac{1}{T-1} \sum_{j=1}^{T-1} \big( R_{T,j} - R_{j,j} \big). \tag{10}$$

**Symbols:** - $T$: total number of stages - $R_{j,j}$: accuracy on task $j$ immediately after it was first learned - $R_{T,j}$: final accuracy on task $j$ after completing all $T$ stages

**Explanation:** BWT measures how the performance on past tasks changes after learning new tasks.

**Meaning:** - If $\text{BWT} \approx 0$, new tasks do not affect old tasks (no forgetting). - If $\text{BWT} < 0$, forgetting occurs. - If $\text{BWT} > 0$, new tasks improve the performance on old tasks, indicating *positive backward transfer*.

$$\text{AvgF} = \frac{1}{T-1} \sum_{j=1}^{T-1} \left( \max_{i \in \{j,...,T\}} R_{i,j} - R_{T,j} \right). \tag{11}$$

**Symbols:** - $T$: total number of stages (or tasks) - $R_{i,j}$: accuracy on task $j$ after training up to stage $i$ - $R_{T,j}$: final accuracy on task $j$ after completing all $T$ stages - $\max_{i \in \{j,...,T\}} R_{i,j}$: the best accuracy achieved on task $j$ during the entire training process

**Explanation:** Average Forgetting (AvgF) measures how much performance is lost on previously learned tasks after completing all training stages. It compares the best performance achieved on each task during training with its final accuracy.

**Meaning:** - AvgF quantifies the extent of catastrophic forgetting across tasks. - A higher AvgF value indicates that the model has forgotten more knowledge from earlier tasks. - An AvgF close to 0 implies strong retention of previously learned knowledge.

$$\text{CARA}_p = \frac{\text{AvgACC} \times \big( 1 - \text{AvgF} \big)}{\text{GFLOPs}^p}, \qquad p \in [0,1]. \tag{12}$$

**Symbols:**
- AvgACC: average accuracy (0–1 scale)
- AvgF: average forgetting (0–1), lower is better
- $1 - \text{AvgF}$: *retention* (forgetting-adjusted performance)
- GFLOPs: inference compute cost (higher means more cost)
- $p$: strength of the compute penalty (typically $0.25 \sim 1.0$; default $p = \frac{1}{2}$)

**Explanation:** Cost-Adjusted Retained Accuracy (CARA) combines *performance* (AvgACC) and *stability* (retention $= 1 - \text{AvgF}$) into a single "retained performance," and normalizes it by a power of the compute cost $\text{GFLOPs}^p$. It summarizes the trade-off among accuracy, forgetting, and efficiency.

**Meaning:**
- Larger values indicate **higher accuracy**, **lower forgetting**, and **reasonable compute** simultaneously.
- $p = \frac{1}{2}$ (*CARA@$\sqrt{cost}$*): applies a mild penalty to compute, balancing performance/stability and efficiency (recommended default).
- $p \to 0$: almost ignores compute (compares retained performance only).
- $p \to 1$: penalizes compute linearly (prioritizes efficiency more).

## C  WHY MEDICAL DATA REQUIRES DIFFERENT PROMPT LEARNING

We argue that, unlike natural images where most continual learning (CL) methods have been studied, medical images possess unique characteristics that necessitate different strategies for prompt learning. To support this claim, we provide a comparative analysis of natural and medical datasets,

Figure 3: To support our claim, we visualized representative examples from the natural images used in prior PCL methods and the medical datasets employed in this study.

as shown in Appendix Figure 3. For natural images, we reference datasets commonly used in prior PCL research Chen et al. (2024); Wang et al. (2022b); Smith et al. (2023); Kim et al. (2024b), including ImageNet Howard et al. (2018), CIFAR100 Krizhevsky (2009), and CORe50 Lomonaco & Maltoni (2017). For medical images, we reference the datasets employed in this work, namely APTOS Karthik et al. (2019), DDR Li et al. (2019), and DRD Dugas et al. (2015).

As illustrated in Appendix Figure 3, medical images are collected under standardized acquisition protocols, resulting in largely consistent capture angles. However, subtle variations emerge due to equipment differences, hospital-specific acquisition procedures, and patient-specific factors. Moreover, accurate diagnosis requires tracking very fine-grained lesions, such as those observed in DDR (Mild) and DRD (Severe). This reinforces our claim that prompt learning in medical CL must be tailored to capture subtle variations in color and lesion details.

In contrast, natural images exhibit substantial intra-class variability in viewpoint, shape, and appearance, even within the same category. Consequently, models trained on natural images must accommodate broad feature differences, which explains why existing PCL methods typically learn wide prompt regions. While this strategy is effective for natural images, it becomes inefficient for medical data, where fundamental anatomical structures remain consistently important across layers and tasks. In independent prompt pools, such low-level features must be redundantly re-encoded at each layer, leading to inefficient use of limited prompt capacity and reducing the number of prompts available for capturing subtle variations. This phenomenon is evident in the red box of Figure 2.a.

To address this limitation, we introduce a unified prompt pool, which enables all layers to share a common set of prompts. This design allows low-level visual prompts, once learned, to be reused at higher layers, thereby avoiding redundant encoding and ensuring efficient utilization of limited prompt resources. As a result, the unified pool not only preserves essential anatomical information but also allocates more capacity toward modeling fine-grained variations, ultimately enhancing the effectiveness of medical continual learning.

The brown points within the red circles in Figure 2.a demonstrate that our proposed prompt effectively models subtle variations. Compared to existing PCL methods, the prompt distribution is significantly more compressed, providing crucial evidence that our proposed prompt learns much finer-grained information than existing methods. Furthermore, prompts learned in this manner consistently demonstrated higher performance across various experiments presented in Tables 1–4. This validates our hypothesis and proves that the integrated prompt pooling method designed based on it contributes to more effective and superior performance on medical data.

# D    MODEL SIZE ANALYSIS: FACEBOOK/DINOV2

Table 9: Parameter counts for facebook/dinov2 models

| Model | Parameters |
|---|---|
| facebook/dinov2-small | 22.1 M |
| facebook/dinov2-base | 86.6 M |
| facebook/dinov2-large | 304 M |
| facebook/dinov2-giant | 1.14 B |

In this section, we explain the rationale for choosing the DINOv2-base model Oquab et al. (2023). In general, models with a larger number of parameters can extract richer features and thus achieve better performance. However, as model size increases, the parameter count grows more than twofold at each step, leading to significantly higher complexity. For this reason, we adopted the moderately sized DINOv2-base model. Remarkably, it still achieved substantially better performance compared to existing methods.

Appendix Table 9 summarizes the four variants of the facebook/dinov2 family in terms of model size and corresponding parameter counts, where M denotes millions of parameters and B denotes billions. These models were employed using the Hugging Face library.

# E    EXPANSION RATIO WE SELECTED

This section presents the experimental results for selecting the prompt expansion ratio. The results are provided in Appendix Table 10, which demonstrate that indiscriminate prompt expansion does not contribute to performance improvement. Based on this insight, we expand the prompts by 100 at each stage, corresponding to 20% of the initial prompts.

Moreover, although we simply expand the prompt set by 100 at each stage, our method consistently outperforms existing PCL approaches that require two rounds of ViT inference. This finding is particularly noteworthy, as it demonstrates that our strategy achieves superior performance with significantly lower computational cost.

Table 10: Performance Evaluation by Prompt Expansion Ratio. ♣ denotes the expansion ratio we selected.

| # Prompt Extensions | Stage (Training Dataset) | APTOS | | DDR | | DRD | |
|---|---|---|---|---|---|---|---|
| | | Acc | F1 | Acc | F1 | Acc | F1 |
| 50 (10%) | Stage 1 (APTOS) | 0.898 | 0.765 | 0.592 | 0.440 | 0.446 | 0.372 |
| | Stage 2 (DDR) | 0.830 | 0.659 | 0.868 | 0.852 | 0.630 | 0.534 |
| | Stage 3 (DRD) | 0.803 | 0.751 | 0.739 | 0.706 | 0.692 | 0.659 |
| **100 (20%)**♣ | Stage 1 (APTOS) | 0.901 | 0.767 | 0.601 | 0.447 | 0.453 | 0.381 |
| | Stage 2 (DDR) | 0.866 | 0.663 | 0.878 | 0.844 | 0.636 | 0.534 |
| | Stage 3 (DRD) | 0.849 | 0.761 | 0.772 | 0.723 | 0.701 | 0.656 |
| 150 (30%) | Stage 1 (APTOS) | 0.901 | 0.774 | 0.608 | 0.469 | 0.470 | 0.408 |
| | Stage 2 (DDR) | 0.816 | 0.668 | 0.869 | 0.856 | 0.637 | 0.543 |
| | Stage 3 (DRD) | 0.836 | 0.757 | 0.757 | 0.709 | 0.694 | 0.636 |

# F    UNIPROMPT-CL TRAINING PROCEDURE

We present Appendix Algorithm 1, UniPrompt-CL Training Procedure (Fixed per-stage prompt expansion), to clarify the training process and the behavior of the model.

---

**Algorithm 1** UniPrompt-CL Training Procedure (Fixed per-stage prompt expansion)

---

**Require:** Initial DINOv2 backbone $\mathcal{M}$; initial prompt keys $K^{(0)}$; initial prompt values $P^{(0)}$; learning rate $\eta$; epochs $E$; batch size $B$; dataset sequence $\{\mathcal{D}_n\}_{n=1}^{N}$; prompt expansion ratio $\rho$ (e.g., 0.2); regularization weight $\lambda_{\text{reg}}$; cosine similarity function $\gamma(\cdot)$; number of dataset $N$; sequence of ViT layer $l$;

**Ensure:** Trained model $\mathcal{M}$ and unified prompt pool $(K, P)$ As the sequence of stages progresses, the model $\mathcal{M}$ is continually updated, i.e., incrementally trained at each stage.

1: $K \leftarrow K^{(0)}, P \leftarrow P^{(0)}$
2: $N_0 \leftarrow |K|, \quad N_{\text{add}} \leftarrow \lfloor N_0 \cdot \rho \rfloor, \quad N_{\text{tot}} \leftarrow N_0$
3: **for** $n = 1$ to $N$ **do**
4: $\quad \mathcal{D} \leftarrow \mathcal{D}_n$
5: $\quad$ **if** $n > 1$ **then**
6: $\quad\quad$ freeze $K[1 : N_{\text{tot}}]$ and $P[1 : N_{\text{tot}}]$
7: $\quad\quad \Delta K \leftarrow \text{RandomInit}(N_{\text{add}}, d); \quad \Delta P \leftarrow \text{RandomInit}(N_{\text{add}}, d)$
8: $\quad\quad K \leftarrow K \cup \Delta K; \quad P \leftarrow P \cup \Delta P$
9: $\quad\quad N_{\text{tot}} \leftarrow N_{\text{tot}} + N_{\text{add}}$
10: $\quad$ **end if**
11: $\quad \text{Opt} \leftarrow \text{AdamW}(\{\theta_{\mathcal{M}}, \theta_{\Delta K}, \theta_{\Delta P}\}, \eta)$
12: $\quad \text{Sched} \leftarrow \text{CosineScheduler}(\text{Opt}, T_{\max} = E)$
13: $\quad$ **for** $e = 1$ to $E$ **do**
14: $\quad\quad$ **for** each mini-batch $(X, Y) \sim \mathcal{D}$ of size $B$ **do**
15: $\quad\quad\quad z_{\text{list}} \leftarrow [\,]$
16: $\quad\quad\quad x \leftarrow X$
17: $\quad\quad\quad p^* \leftarrow \text{NormalizeBatchPrompt}(P)$
18: $\quad\quad\quad$ **for** $\ell = 1$ to $l_{\text{use}}$ **do**
19: $\quad\quad\quad\quad x \leftarrow \mathcal{M}_{\ell}(x)$
20: $\quad\quad\quad\quad q_{\ell} \leftarrow \text{CLS}(x)$
21: $\quad\quad\quad\quad \phi_{\ell} \leftarrow \gamma(q_{\ell} K)\, P$
22: $\quad\quad\quad\quad x_{\text{next}} \leftarrow f_{\ell}(x, \phi_{\ell})$
23: $\quad\quad\quad\quad x \leftarrow x_{\text{next}}$
24: $\quad\quad\quad\quad z_{\ell} \leftarrow \text{Softmax}\left( \dfrac{K\,(p^*)^{\top}}{\|K\|\,\|p^*\|} \right)$
25: $\quad\quad\quad\quad$ append $z_{\ell}$ to $z_{\text{list}}$
26: $\quad\quad\quad$ **end for**
27: $\quad\quad\quad$ **for** $\ell = l_{\text{use}} + 1$ to $l_{\text{total}}$ **do**
28: $\quad\quad\quad\quad x \leftarrow \mathcal{M}_{\ell}(x)$
29: $\quad\quad\quad$ **end for**
30: $\quad\quad\quad \hat{Y} \leftarrow \text{Classifier}(x)$
31: $\quad\quad\quad \mathcal{L}_s \leftarrow \mho(z_{\text{list}})$ (defined in Eq. (6))
32: $\quad\quad\quad \mathcal{L}_{\text{CE}} \leftarrow \text{CrossEntropy}(\hat{Y}, Y)$
33: $\quad\quad\quad \mathcal{L}_{\text{total}} \leftarrow \mathcal{L}_{\text{CE}} + \lambda_{\text{reg}} \mathcal{L}_s$
34: $\quad\quad\quad$ Backpropagate $\nabla \mathcal{L}_{\text{total}}$; Opt.step()
35: $\quad\quad$ **end for**
36: $\quad\quad$ Sched.step()
37: $\quad$ **end for**
38: $\quad \text{Eval}(n) \leftarrow \text{Evaluate\_Tests}(\mathcal{M}, \{\mathcal{D}_n\}_{n=1}^{N})$
39: **end for**
40: **return** $\mathcal{M}, (K, P)$

---

## G  KEY HYPERPARAMETERS FOR BASELINES

To further strengthen the reproducibility of our experiments, we include Appendix Table 11, which lists the key hyperparameters for the prompt-based continual learning baselines. Whenever these CL methods are used as baselines, we train them using the hyperparameters specified in this table. Consistent with prior practice, we primarily adopt the default settings from the rigorously implemented CL framework Boschini et al. (2022); Buzzega et al. (2020). When a method's default produced

Table 11: Key hyperparameters for prompt-based continual learning baselines.

| Method | Hyperparameter | Value |
|---|---|---|
| CODA-Prompt | Prompt pool size (`coda_pool_size`) | 100 |
| | Prompt length (`coda_prompt_length`) | 4 |
| | Ortho. regularization coeff. (`coda_ortho_mu`) | 0.1 |
| L2P | Prompt length (`l2p_prompt_length`) | 4 |
| | Prompt pool size (`l2p_pool_size`) | 100 |
| | Top-$k$ prompts (`l2p_top_k`) | 5 |
| | Head type (`l2p_head_type`) | prompt |
| | Embedding key (`l2p_embedding_key`) | cls |
| DualPrompt | Prompt length (`dual_prompt_length`) | 4 |
| | Prompt pool size (`dual_pool_size`) | 100 |
| | Top-$k$ prompts (`dual_top_k`) | 5 |
| | G-prompt length (`dual_g_prompt_length`) | 4 |
| | G-prompt layers (`dual_g_prompt_layers`) | [0, 1] |
| | E-prompt layers (`dual_e_prompt_layers`) | [2, 3, 4] |
| | Head type (`dual_head_type`) | token |
| DER++ | ALPHA (`larger values emphasize preserving past knowledge`) | 0.5 |
| | BETA (`larger values emphasize relearning past tasks`) | 0.5 |
| | BUFFER_SIZE (`number of stored past samples`) | 30 |
| Online EWC | RP_SIZE (`dimension of the random projection matrix`) | 10000 |
| | OPTIM_MOM (`SGD momentum`) | 0.9 |
| | OPTIM_WD ($\ell_2$ `regularization coefficient`) | 0.0005 |
| MoE Adapters | PROMPT_TEMPLATE (`CLIP text prompt template`) | "a photo of {}." |
| | CLIP_BACKBONE (`CLIP model backbone`) | ViTB/16 |
| | VIRTUAL_BS_N (`virtual batch size iterations`) | 1 |

unreasonably poor performance, we adjusted the hyperparameters to ensure a fair comparison and reported the best validated configuration (explicitly recorded in Appendix Table 11).

## H DATASETS FOR SMALL-SCALE EXTERNAL EXPERIMENTS

We conducted supplementary small-scale external experiments to assess generalization beyond DR. Specifically, we constructed a domain-incremental scenario using three skin-cancer datasets with undersampling. For all datasets, the original diagnostic categories were remapped into a clinically motivated binary setting, following prior work on ISIC lesion grouping Cassidy et al. (2022): lesions that require urgent clinical intervention were grouped as *malignant*, whereas comparatively less urgent lesions were grouped as *benign*. Dataset distributions are provided in Appendix Tables 12, and representative example images from the three datasets are shown in Appendix Figure 4. All data were split 8:1:1 into train, validation, and test sets. The training order is ISIC → HAM → DERM7, yielding three stages. All other settings (model, training schedule, hyperparameters, and metrics) are identical to the DR study—only the datasets change, and the batch size is set to 16 to account for the smaller dataset size. For reproducibility, we include the dataset split CSVs in the released code, consistent with the DR setup.

Table 12: Distribution of datasets used for training

| Dataset | benign | Malignant |
|---|---|---|
| ISIC Rotemberg et al. (2021) | 584 | 584 |
| HAM Codella et al. (2019) | 1113 | 1113 |
| DERM7 Kawahara et al. (2018) | 252 | 252 |

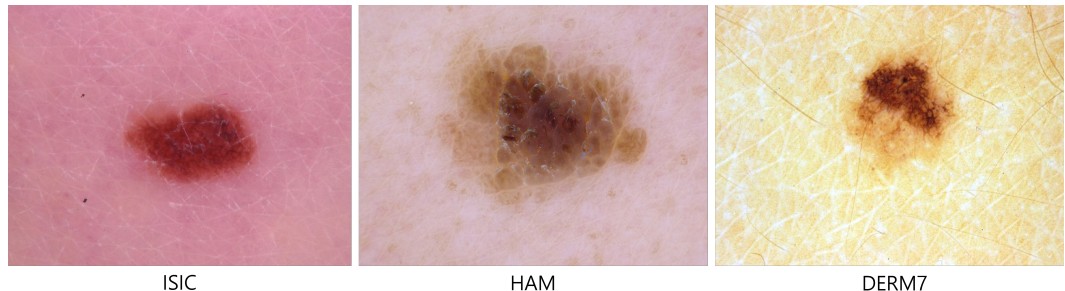

ISIC          HAM          DERM7

Figure 4: Representative examples from the three skin cancer datasets used in our pilot study: ISIC Rotemberg et al. (2021), HAM Codella et al. (2019), and DERM7 Kawahara et al. (2018). The samples illustrate the diversity in lesion appearance and acquisition conditions across datasets.

# I  GENAI USAGE DISCLOSURE

Although the authors carried out the overall code development and manuscript preparation, the conversion of the written manuscript into LaTeX, its proofreading, and translation were performed using OpenAI's ChatGPT, and code refactoring was assisted by Cursor.

