# OpenReview forum: "UniPrompt-CL: Sustainable Continual Learning in Medical AI with Unified Prompt Pools"
_ICLR.cc/2026/Conference — Submitted to ICLR 2026_

### Official Review · Reviewer_Ab34 · 2025-10-28

**Soundness:** 3
**Presentation:** 3
**Contribution:** 2
**Rating:** 2
**Confidence:** 3

**Summary:**

This paper introduces a continual learning approach with an application in learning from images from different hospitals (in particular ophthalmology). The paper is easy to read and the method is motivated well. Some good improvements are shown especially over the method by Kim.

**Strengths:**

- For the specific setting discussed (multiple hospitals that cannot share data) the methods are suited
- Methods are clearly described
- There is an improvement over the existing methods for this specific task

**Weaknesses:**

- The paper is very incremental from Kim 2025 (or 2024?).
- Although the paper aims to be quite general in several parts of the descriptions the motivation for doing this is quite specific and also the experiments are focused on a very specific application and dataset.  This would be suited for a a medically oriented conference but for a venue as ICLR you want methods to generalize beyond the specific application. That might be possible in this case but that point cannot be convincingly made with the current paper.

Minor things:
- In table 4 there is a mismatch between the ECCV2024 and the statement after that (2025).
- Title talks about sustainability but that is not something that is really addressed in the paper

**Questions:**

- What are the precise technical extensions made to the method by Kim?
- What other applications do you envision for the proposed methods?

**Details Of Ethics Concerns:**

N.A.

---

> ### Author Response · Authors · 2025-11-19
> **Response to Reviewer Ab34**
>
> We thank Reviewer **Ab34** for the constructive suggestions.
> We are committed to addressing your concerns as thoroughly as possible and summarize our responses below.
>
> ### 1. What technical extensions does the proposed method introduce?
>
> The OS-Prompt(++) literature explicitly reports a **significant degradation in performance when no additional query module (query function) is used**. In other words, existing PCL architectures are heavily dependent on a separate query network, making it difficult to avoid a **trade-off between model simplicity / inference efficiency and performance**. To address this limitation, and motivated by the observations in Figure 2(a) and **Appendix Figures 3 and 4** that medical images are acquired under standardized imaging protocols and that **precise capture of fine signals** such as blood vessels, drusen, inflammation, and lesions is crucial, we propose a Prompt-CL method specialized for the medical domain. Our technical contributions can be summarized in three main aspects:
>
> 1. **Unified prompt pool (Shared Unified Pool)**
>    Similar to OS-style methods, we use the \([{\rm CLS}]\) token as a query and construct prompts via cosine-similarity-based continuous weighted combinations. However, whereas OS-Prompt(++) maintains separate prompt pools for each layer—preventing prompts from interacting across layers—we design a **single unified pool shared by all layers**, enabling global reuse and information exchange of prompts across the network. As a result, each layer’s query can refer to accumulated global information, achieving **more stable and powerful selection without additional query/selection modules**, and in practice, outperforming OS-style baselines.
>
> 2. **Fixed per-stage expansion**
>    We freeze the initial prompts and, at each CL stage, linearly add only a small, fixed ratio of new prompts. These newly added prompts are shared across all layers through the unified pool, which encourages learning **global continuity** among queries generated at different layers. In addition, we explicitly encourage the expanded prompts to capture information that minimally overlaps with the existing prompts, thereby promoting greater diversity in the prompt set. The benefits of this design are supported by the clear performance gains observed in Figure 2(b) and multiple quantitative and qualitative experiments.
>
> 3. **Regularizer for encouraging learning of new prompts**
>    At early expansion stages, the influence of newly added prompts can be relatively weak, which may lead to **gradient dilution** and limited learning contributions. To address this, we introduce a regularization term (Eq. (6)) that encourages new prompts to interact with existing ones and learn **additional, complementary task-related information**. This yields both stable and efficient prompt learning. The importance of this component is validated by ablation experiments in Table 5, which show performance degradation when the regularizer is removed.
>
> In summary, the proposed method achieves **stable, reusable, and efficient** prompt selection **without an additional query module**. By allowing queries at all layers to refer to a **shared global prompt representation** under a single unified pool, our framework provides a learning structure that is specifically tailored to **fine-grained cues** crucial in the medical domain, making it clearly distinct from OS-style and related methods.
>
> ### 2. What other applications do you envision for the proposed method?
>
> Based on additional small-scale domain-incremental experiments using DR and three skin cancer datasets, where the proposed method outperforms prior approaches in terms of AvgACC, we partially demonstrate its **transferability across medical imaging tasks**. Accordingly, we believe the method can be extended to a variety of medical applications, including:
>
> - **3D CT/MRI segmentation**
> - **Lesion detection and classification in chest X-ray**
> - **Ultrasound lesion analysis**
> - **Pathology (WSI) tile classification and detection**, among others.
>
> Furthermore, our approach has advantages in settings where data sharing is restricted. The combination of a **single unified prompt pool** with a strategy of **freezing prompts and adding only a small, fixed ratio of new prompts** at each stage enables **stable adaptation with minimal additional computation** when hospitals, vendors, or domain change. The **single-pass inference** structure is also beneficial in clinical deployment scenarios with strict **latency and resource constraints**. **Therefore, we expect the method to be highly applicable in constrained data environments at both institutional and national scales.**

---

> > ### Comment · Reviewer_Ab34 · 2025-11-26
> >
> > Thanks for the clarification, and good to see some additional datasets being considered. For me the method remains more suited for a specific medical imaging venue than ICLR and my statement on being incremental over Kim remains.

---

> ### Author Response · Authors · 2025-11-28
> **Response to Reviewer Ab34 (request for clarification).**
>
> **Thank you for the follow-up, Reviewer Ab34. We appreciate your time and thoughtful feedback.**
>
> To ensure we address your concern precisely, could you please clarify which metrics, settings, or comparisons you had in mind when concluding that the improvement is limited?
>
> From our perspective, the contribution goes beyond a venue-specific tweak and is supported by structural changes validated through causal ablations:
> (1) a single, layer-shared prompt pool (no additional query module),
> (2) fixed-ratio few-prompt expansion per stage, yielding predictable compute growth,
> (3) a regularization term that increases the utilization of newly added prompts.
> These components show additive gains in our ablation studies.
>
> Empirically, we also broadened validation beyond DR by conducting domain-incremental experiments on three skin-cancer datasets and comparing against non-PCL families (rehearsal-, regularization-, and architecture-based methods). Moreover, our CARA metric indicates higher retained accuracy per unit compute, highlighting practical relevance under latency and resource constraints.
>
> Finally, we acknowledge that our current design is primarily motivated by settings with consistent imaging protocols (as in many clinical workflows). However, such protocol consistency is not unique to medical imaging. Similar characteristics appear in industrial anomaly detection benchmarks such as MVTec[1] and BeanTech[2]. Considering how rapidly industrial environments evolve, we believe methods that can continually adapt under such conditions may become even more necessary.
>
> **We truly value your feedback, and if these points alleviate the concern, we would appreciate your reconsideration of the paper’s contribution and practical impact.**
>
> [1] Paul Bergmann, Kilian Batzner, Michael Fauser, David Sattlegger, Carsten Steger: The MVTec Anomaly Detection Dataset: A Comprehensive Real-World Dataset for Unsupervised Anomaly Detection; in: International Journal of Computer Vision 129(4):1038-1059, 2021, DOI: 10.1007/s11263-020-01400-4.
> [2] Mishra, Pankaj, et al. "VT-ADL: A vision transformer network for image anomaly detection and localization." 2021 IEEE 30th International Symposium on Industrial Electronics (ISIE). IEEE, 2021.

---

### Official Review · Reviewer_XbYJ · 2025-10-30

**Soundness:** 1
**Presentation:** 2
**Contribution:** 2
**Rating:** 2
**Confidence:** 4

**Summary:**

This paper identifies the suboptimal performance and high computational cost of existing prompt-based continual learning (PCL) methods when applied to the medical domain. To address these limitations, the authors propose **UniPrompt-CL**, a novel framework specifically designed for continual learning in healthcare.

The core of this method is a unified prompt pool, where prompts are shared across all layers of the model. At each layer, a query derived from the [CLS] token is used to select and combine prompts from this shared pool to generate a layer-specific instruction. This design is motivated by the authors' hypothesis that medical images necessitate a focus on capturing fine-grained details, as opposed to the more varied and broadly dispersed features found in natural images.

Consequently, the proposed method achieves state-of-the-art (SOTA) performance on medical CL benchmarks while significantly improving computational efficiency by requiring only a single inference pass, unlike prior approaches that often necessitate two.

**Strengths:**

1.  **Addresses a Significant Problem with a Well-Reasoned Motivation**
    * The paper is grounded in a compelling and well-reasoned rationale: that medical images require a fundamentally different continual learning strategy than natural images.
    * The motivation is built on the clear distinction between the standardized nature and fine-grained analytical needs of medical data versus the high variability of natural images, providing a strong foundation for the work.
***
2.  **Proposes a Useful Conceptual Framework for Domain-Specific PCL**
    * The work introduces a valuable **"dispersed vs. clustered" framework** to analyze and explain the behavior of prompts on different data domains.
    * This conceptual lens provides a useful and insightful perspective for the research community to evaluate and design future PCL methods tailored for specialized domains.
***
3.  **Demonstrates Clear and Quantifiable Computational Efficiency Gains**
    * The proposed method has a clear practical advantage by achieving superior performance with only a **single inference pass**, directly addressing a major limitation of prior SOTA methods.
    * This efficiency gain is explicitly quantified, resulting in a **~33% reduction in FLOPs** compared to the top-performing baseline (44.17 GFLOPs vs. 66.42 GFLOPs), as shown in Table 2. This is a significant improvement for real-world applications.

**Weaknesses:**

1.  **Lack of Clarity in Methodology Compromises Reproducibility**
    * The manuscript's most significant weakness is the ambiguous and incomplete description of the training procedure, which hinders the reader's ability to understand and reproduce the work. Specifically, the procedure for training the model on each successive continual learning (CL) stage is ambiguous.
    * The description in Section 4.2 is particularly confusing and logically contradictory: *"we first construct and train the integrated prompt pool as described in Section 4.1, and then freeze the prompt weights corresponding to the previous stages"*. This implies that prompts are trained *before* being frozen, which is counter-intuitive for a CL process. It is unclear if each new task involves one or two distinct training phases.
    * Furthermore, key implementation details for baselines are missing. To fairly assess the proposed method's parameter efficiency, the number of learnable parameters for the compared baseline methods should be reported.

***

2.  **Insufficient Empirical Support for the Central Causal Claim**
    * The paper's central claim is that the architectural design of the **"unified prompt pool"** is the inherent cause for learning the **"clustered"** prompt representations supposedly ideal for medical data, as shown in Figure 2.a.
    * However, this causal link is not rigorously substantiated. The observed prompt distribution could be a result of **confounding variables**, such as specific hyperparameter choices (e.g., learning rate, optimizer, regularization strength), rather than a direct and inevitable consequence of the architectural design itself. The paper does not provide experiments to decouple these effects.


***

3.  **Limited Experimental Scope Restricts Generalizability Claims**
    * The authors frame their method as a framework for "healthcare" and "medical AI", suggesting broad applicability.
    * However, the empirical validation is confined to a **single task (classification) on a single modality (fundus photography for diabetic retinopathy)** across three similar datasets. This narrow experimental scope does not sufficiently support the broad claims of generalizability across the medical domain. The authors themselves acknowledge this limitation in the future work section.

***

4. **Inconsistent Reporting of Core Continual Learning Metrics**
    * The manuscript's treatment of core continual learning (CL) metrics lacks internal consistency. Given that the work addresses catastrophic forgetting, the direct measurement of this phenomenon is a critical component of the evaluation.
    * In Section 5, the authors explicitly state their decision to omit standard forgetting metrics like Backward Transfer (BWT) and Average Forgetting (AvgF), citing the "limited-size medical datasets" as the reason.
    * This statement is directly contradicted in a later part of the paper. In Section 7 and Table 4, the authors present and analyze results for these exact metrics (BWT and AvgF) as part of their ablation study.
    * Such a direct contradiction between different sections of the paper affects the clarity and perceived rigor of the experimental evaluation.

**Questions:**

#### **Regarding the Methodology**

1. **On the Training Procedure for Each CL Stage:**
    The current description of the training process is ambiguous, which may pose a challenge for reproducibility.
    * Could the authors please provide a detailed, step-by-step description or, ideally, an algorithmic pseudo-code of the training procedure for a single continual learning (CL) stage?
    * Specifically, clarification is requested for the statement in Section 4.2: *"we first construct and train the integrated prompt pool... and then freeze the prompt weights corresponding to the previous stages."* Could the authors clarify if this implies a two-phase training process for each new task, and elaborate on the exact sequence of operations?

2. **On the Prompt Pool Design and Comparison to Prior Art:**
    The 'unified prompt pool' is presented as a key innovation. However, its relationship with similar concepts in prior work could be further clarified.
    * Methods like Coda-Prompt, L2P and DualPrompt also appear to use a form of unified, layer-shared prompt pool. Could the authors first elaborate on the key architectural and methodological differences between UniPrompt-CL and Coda-Prompt?

3.  **On the Substantiation of the "Dispersed vs. Clustered" Claim:**
    The paper's motivation hinges on the claim that baseline methods learn "dispersed" prompts, while the proposed method learns "clustered" ones.
    * Was this conclusion based on a systematic analysis across various experimental conditions, or primarily on the visual observation presented in Figure 2.a?
    * More importantly, how did the authors ensure that this observed difference is a direct consequence of the architectural design (unified vs. independent pools) rather than an artifact of hyperparameter settings? For instance, could the "dispersed" nature of baseline prompts be altered if their hyperparameters were tuned specifically for the medical datasets? A more thorough analysis to decouple these effects would greatly strengthen this central claim.

***

#### **Regarding the Experiments**

4.  **On the Scope of Generalizability Claims:**
    The paper suggests the method is designed broadly for "healthcare" and "medical AI."
    * Given that the experiments are conducted exclusively on diabetic retinopathy classification, could the authors comment on the method's expected generalizability to other medical tasks (e.g., segmentation, detection) and modalities (e.g., CT, MRI)? This would help to properly contextualize the scope of the contributions.

5.  **On the Comprehensive Evaluation of Catastrophic Forgetting:**
    Catastrophic forgetting is a critical metric in any continual learning evaluation.
    * The main results in Table 1 report final accuracy, while forgetting metrics (BWT, AvgF) are only provided for a subset of methods in the ablation study. To provide a more comprehensive and necessary comparison of the method's SOTA performance in mitigating forgetting, would it be possible for the authors to provide the BWT and AvgF results for **all** baseline methods evaluated in Table 1?

6.  **On the Efficiency Comparison with All Baselines:**
    The paper rightly emphasizes computational efficiency as a key advantage.
    * To allow for a complete assessment, could the authors provide a table comparing both the **inference FLOPs** and the **total number of learnable parameters** for **all** baseline methods listed in Table 1, not just OS-Prompt++?

7.  **On Data Processing for Reproducibility:**
    * Could the authors please specify in the appendix how the original 5 classes of the APTOS dataset were mapped to the 3 classes used in the experiments? This detail would greatly aid in the full reproducibility of the results.

---

> ### Author Response · Authors · 2025-11-19
>
> We are grateful to Reviewer **XbYJ** for the highly constructive comments.
>
> **We are committed to addressing all concerns raised in the review as thoroughly as possible.**
> Below, we provide our detailed responses.
>
> ### 1. regarding the training procedure at each CL stage
>
> While the existing training procedure is briefly described in Section 5 (Experimental Details), we agree that a more detailed explanation would be beneficial, as pointed out by the reviewer. In response, we have added **Appendix F, *UniPrompt-CL Training Procedure***, where we present the stage-wise training flow and model behavior (prompt freezing and stage-wise fixed-ratio expansion) in pseudocode form. We have also updated Section 5 of the main text to reflect these details. We hope these clarifications address the reviewer’s concerns.
>
> ### 2. Prompt pool design and comparison with prior work
>
> The high-level differences between our method and existing work are outlined in the PRELIMINARY section. Here, we provide a more concrete discussion of how our approach relates to prior PCL methods.
>
> Coda-Prompt, L2P, and DualPrompt all rely on prompt pools (or shared pools plus a selection mechanism) that can be used across layers, and in that sense they are conceptually related to our approach. However, their core designs differ significantly from ours. These methods typically employ a combination of a global shared pool and task-specific pools, and require additional query functions. The prompt selection mechanisms also vary across methods.
>
> In contrast, we do not maintain separate shared and task-specific pools in parallel; instead, we use a **single unified pool** only. While we also **reuse the [CLS] token as a query** and construct prompts via cosine-similarity-based **continuous weighted combinations**, similar to OS-style methods, our design differs in that the prompt pool is shared across all layers and learns the **global continuity among queries** generated at different layers. Importantly, we do **not** rely on **discrete top-\(K\) selection** strategies, nor do we require the **additional query/selection modules** used in methods such as Coda-Prompt.
>
> As training progresses through the stages, our method **linearly adds only a small, fixed ratio of new prompts** at each stage while **freezing existing prompts**, thereby strictly controlling the growth in computational cost and parameter count. In addition, by introducing a regularization term (Eq. (6)), we encourage new prompts to interact with existing ones and learn **complementary task-specific information**, achieving both stable and efficient prompt learning. This design allows us to surpass prior methods **without adding an extra query network**, i.e., with **only a single ViT forward pass at inference time**.
>
> In summary, UniPrompt-CL:
>
> - Achieves **high reusability via a single unified pool**,
> - Maintains **structural simplicity by minimizing unnecessary modules**, and
> - Strictly controls the growth of prompt count and computational cost, ensuring **efficiency**.
>
> This distinguishes our framework at the design level from prior approaches that rely on shared + task-specific pools, discrete top-\(K\) selection, or separate query networks.

---

> > ### Author Response · Authors · 2025-11-19
> >
> > ### 3. Evidence for the “dispersed vs. clustered” prompt distribution argument
> >
> > The motivation for our work originates from the observations in Figure 2(a) and **Appendix Figures 3 and 4**. Specifically, by comparing natural images with medical images (DR and skin cancer) in Appendix Figure 3 and the newly added Figure 4, we observed that medical data are generally acquired under **standardized imaging protocols** and thus exhibit **similar viewpoints and framing**. As discussed in Appendix C, this implies that in the medical domain, the primary challenge is not distinguishing large shape variations, but rather **accurately capturing fine signals** such as blood vessels, drusen, inflammation, and lesions (our first motivation).
> >
> > Building on this observation, we visualized the prompt distributions using OS-Prompt++ as in Figure 2(a), and found that existing PCL methods tend to produce **overly dispersed prompts across layers**, often **duplicating similar features**.
> >
> > We thus hypothesized that PCL methods optimized for natural images are designed to be **sensitive to coarse (large-scale) variations** such as viewpoint, texture, and style, whereas medical data require strategies that consistently amplify **fine-grained cues**.
> >
> > To address this, we propose a unified prompt pool that **linearly expands by a small, fixed ratio at each stage**, combined with a regularization term (Eq. (6)) that **encourages meaningful learning of new prompts**. As shown in the main text and appendix, our method achieves superior performance and efficiency compared to various alternatives, thereby **empirically supporting** the above hypothesis.
> >
> > ### 4. On the scope of our generalization claims
> >
> > In conclusion, we believe that the proposed framework has strong potential to generalize to medical imaging domains that follow standardized acquisition protocols (e.g., CT, MRI, X-ray). This claim is supported by Appendix Figure 3, the newly added Figure 4, and the additional small-scale domain experiments.
> >
> > First, Appendix Figures 3 and 4 show that, compared to natural images, medical images are typically acquired under well-defined imaging protocols, resulting in similar viewpoints and framing. As discussed in Appendix C, performance in the medical domain hinges less on distinguishing large shape variations and more on how accurately one can capture fine-grained signals such as blood vessels, drusen, inflammation, and lesions. This directly supports the design motivation of our method and extends naturally to skin cancer data.
> >
> > Second, beyond the DR scenario presented in the main text, we conducted additional small-scale domain-incremental experiments using three skin-cancer datasets, and the results are reported in Table 6. The proposed method outperforms prior approaches in terms of AvgACC, empirically demonstrating its potential to generalize **across diseases and organs**. Combined with the quantitative comparisons of performance and efficiency in the main text and appendix, these results suggest that our method can be effectively applied and extended to a wide range of medical imaging domains with standardized acquisition protocols.
> >
> > ### 5. Comprehensive evaluation of catastrophic forgetting and efficiency across all baselines
> >
> > We group together several related questions here. In response to the reviewer’s request, we computed **forgetting metrics** (e.g., BWT, AvgF) for all baselines, and also introduced metrics based on **FLOPs** and **performance per compute**. These results are summarized in Table 4 and Table 6. Additionally, we provide a detailed description of the Cost-Adjusted Retained Accuracy (CARA) metric in Appendix B, and include the corresponding results and analysis in Section 7, *“Quantitative validation using forgetting metrics”*.
> >
> > ### 6. Data processing for reproducibility
> >
> > Following criteria proposed by ophthalmologists [1] , we consistently remap the original 5 classes (normal, mild DR, moderate DR, severe DR, proliferative DR) into 3 classes (normal, non-proliferative DR (NPDR), proliferative DR (PDR)). Concretely, *mild, moderate, and severe DR* are merged into a single *NPDR* class, while the original normal and PDR labels are preserved as is.
> >
> > To further support reproducibility, we provide a link in the abstract that grants access to all relevant materials, including training code, model code, and data splits. This information is also reinforced in Appendix A. If permitted under ICLR policy, we plan to update the same link with code and splits for the newly added experiments; if policy constraints arise, we will incorporate these updates at the camera-ready stage instead.
> >
> > [1] Sabiha Gungor Kobat, Nursena Baygin, Elif Yusufoglu, Mehmet Baygin, Prabal Datta Barua, Sengul Dogan, Orhan Yaman, Ulku Celiker, Hakan Yildirim, Ru-San Tan, et al. Automated diabetic retinopathy detection using horizontal and vertical patch division-based pre-trained densenet with digital fundus images. Diagnostics, 12(8):1975, 2022.

---

### Official Review · Reviewer_zHwv · 2025-10-30

**Soundness:** 2
**Presentation:** 3
**Contribution:** 2
**Rating:** 4
**Confidence:** 4

**Summary:**

This paper proposes UniPrompt-CL, a framework designed to tackle catastrophic forgetting in medical AI by combining a strong backbone with efficient prompt learning. The method is evaluated on three diabetic retinopathy datasets, outperforming existing state-of-the-art PCL methods. The work also highlights the unique challenges posed by medical data, as their standardized nature impacts the effectiveness of traditional PCL methods. By incorporating an integrated prompt pool and lightweight expansion, UniPrompt-CL effectively preserves prior knowledge and captures fine-grained features, ensuring robust and stable performance.

**Strengths:**

1. A reasonable method is proposed to tackle an important problem, i.e., prompt-based continual learning in medical domain.
2. The experimental results show clear performance improvements over several prompt-based CL methods.
3. The writing is generally well.

**Weaknesses:**

1. The paper states that standard Prompt-based Continual Learning methods fail on medical data because they are designed for the broad feature space of natural images, whereas medical images require more fine-grained distinctions. However, this claim is supported by limited and largely qualitative evidence, primarily a single t-SNE visualization.  The analysis lacks depth and fails to investigate the fundamental reasons for this performance gap.

2. The comparison lacks other CL families (e.g., regularization, rehearsal, LoRA-based, etc).

3. The paper's title and abstract make broad claims about advancing "Medical AI," yet the experiments are confined to diabetic retinopathy datasets.

**Questions:**

Please see the weakness.

---

> ### Author Response · Authors · 2025-11-19
> **Response to Reviewer zHwv**
>
> First, we sincerely thank Reviewer **zHwv** for providing additional references and detailed, constructive feedback. We have carefully examined the concerns you raised and are committed to revising the paper as thoroughly as possible to address them.
>
> ### 1. Why existing PCL methods do not work well in the medical domain
>
> Our central hypothesis is that *“standard PCL architectures designed for natural images inherently favor ‘broad and generic’ features, and therefore have structural limitations in capturing the ‘narrow and fine-grained’ diagnostic signals of medical images.”* This hypothesis is supported by the following quantitative and qualitative results.
>
> First, in Table 1 and Table 4, existing SOTA PCL methods underperform UniPrompt-CL by at least **10 percentage points in accuracy and 9 points in F1-score** across all DR medical datasets (APTOS, DDR, DRD), and, as can be seen from the standard CL metrics (e.g., AvgACC, BWT, AvgF), they also suffer substantially larger degradation in continual-learning performance. Such a performance gap is difficult to explain purely by hyperparameter differences and instead serves as strong quantitative evidence that existing PCL methods fail to adequately capture the signal structure characteristic of medical data. If the standard PCL design were truly well-suited for medical images, it would be unlikely to observe such large and consistently repeated performance gaps.
>
> We further hypothesize that this quantitative gap is not a coincidence but a consequence of a **mismatch in design philosophy**, and we investigate this through analyses of prompt distributions and domain characteristics. In the t-SNE visualization in Figure 2(a), the prompts of OS-Prompt are **widely and loosely dispersed** across the feature space, whereas the unified prompt pool of UniPrompt-CL forms **dense and compact clusters**. As shown in Appendix C and Figure 3, natural images exhibit **extreme intra-class variability** due to changes in viewpoint, background, and illumination, whereas medical images rely on **subtle variations in color, texture, and local lesions** on top of a fixed anatomical structure. A standard PCL design that aims to cover a wide feature space therefore risks diffusing crucial fine-grained medical signals as noise in this broader representation space, which in turn leads to **reduced discriminative power → performance gaps**, in line with our interpretation.
>
> In contrast, UniPrompt-CL employs a **unified prompt pool** that shares and reuses prompts learned at lower layers, directing the limited prompt capacity away from “wide coverage” and towards **medical-specific, local features**. The fact that we observe similar trends not only on DR datasets but also in the small-scale skin cancer domain-incremental experiments reported in the paper suggests that this hypothesis is not restricted to a particular dataset, but may reflect a **structural pattern common across medical imaging**.
>
> In summary, by jointly considering **domain characteristics (broad variability vs. fine-grained changes), prompt distribution patterns, and consistent performance gaps**, we hypothesize that standard PCL is structurally disadvantaged for medical images. We then propose a method to address this issue and validate it through extensive quantitative and qualitative experiments.
>
> ### 2. Additional PCL methods and dataset extensions
>
> This concern is closely related to that of Reviewer **uosG**, and we therefore summarize the corresponding additions here.
>
> To address the concern that restricting the main experiments to DR datasets may weaken the **generality claim** of the framework, and that **a broader set of baseline models** is needed, we made the following revisions. We conducted additional **small-scale domain-incremental experiments** using three public skin cancer datasets and expanded the baselines to include **prompt-based CL**, **regularization-based CL**, **rehearsal-based CL**, and **architecture-based CL** methods. The corresponding configurations are reflected in the updated Tables 1, 4, and 6, and the dataset splits and implementation details are described **in detail in Appendix H** to further strengthen reproducibility.
>
> As a result, even in the skin cancer domain-incremental setting, the proposed method **still achieves the highest AvgACC**, confirming that the framework can be extended **beyond DR to other diseases and imaging domains**. Furthermore, using the CARA (Cost-Adjusted Retained Accuracy) metric, which jointly accounts for computational cost, we quantitatively show that our method delivers higher practical utility (accuracy per compute) compared to baselines. Specifically, the proposed method achieves \( CARA_{0.5} = 0.105 \), surpassing strong baselines such as OS (0.101), OS++ (0.095), DER++ (0.050), Coda-Prompt (0.059), and MoE-Adapters (0.056), thereby supporting our claim that it **retains and delivers more accuracy under the same or similar compute budget**.

---

> > ### Author Response · Authors · 2025-11-19
> >
> > Here, **CARA** combines AvgACC and a GFLOPs-based penalty term to **summarize the trade-off between accuracy and compute into a single scalar**. The **exact formulation, computation procedure, and interpretation** are detailed in Appendix B.

---

> > > ### Comment · Reviewer_zHwv · 2025-11-27
> > >
> > > Thanks for the responses. The added experiments on more datasets and comparisons with CL-related methods addressed my concerns. Therefore, I have raised my rating.

---

> > > > ### Author Response · Authors · 2025-11-28
> > > > **Thank You for the Re-evaluation ( zHwv )**
> > > >
> > > > We sincerely appreciate your follow-up and the updated rating. It’s great to hear that the extended experiments and the CL-family comparisons addressed your concerns. If any other analyses or pointers would be useful for the camera-ready, we’re happy to incorporate them.

---

### Official Review · Reviewer_uosG · 2025-11-05

**Soundness:** 3
**Presentation:** 2
**Contribution:** 2
**Rating:** 6
**Confidence:** 4

**Summary:**

This paper proposes a new prompt-based continual learning method termed UniPrompt-CL, motivated by data sharing constraints across institutions in medical imaging settings. The method uses a unified prompt pool that shares prompts across all transformer layers, combined with minimal prompt expansion and a new regularization term. They empirically evaluate their method on a sequence of diabetic retinopathy tasks (APTOS, DDR, DRD) in the domain-incremental learning setting, demonstrating that they outperform previous prompt-based approaches for the considered tasks.

**Strengths:**

1. Clear motivation. The paper articulates well why CL techniques are critical for medical settings under data-sharing restrictions and how current PCL frameworks may fail in this domain.
2. The method is computationally efficient compared to dual-inference methods.
3. The method outperforms the considered PCL baselines for the considered diabetic retinopathy task.

**Weaknesses:**

1. The main weakness is limited empirical evaluation. The paper only conducts evaluation on one benchmark. To strengthen the empirical claims, the authors should consider evaluating the method on more tasks in the medical domain and ideally for longer sequences (more than 3 domains).  [1] Table 6 and [2] DermCL contain tasks in the domain incremental learning setting, which are publicly available.
2.  The main motivation of this work is for continual learning methods in medical settings. However, it lacks discussion of current continual learning methods in the medical domain. [3][4][5] specifically account for the domain incremental setting.

[1] Continual Learning in Medical Image Analysis: A Comprehensive Review [2] Expert Routing with Synthetic Data for Continual Learning [3] Feature Transformers: Privacy Preserving Lifelong Learners for Medical Imaging. [4] Conditional diffusion replay for continual learning in medical settings. [5] Generalizable Continual Classification of Medical Images.

**Questions:**

See weaknesses above.

---

> ### Author Response · Authors · 2025-11-19
> **Response to Reviewer uosG**
>
> First, we sincerely thank Reviewer **uosG** for providing additional references and detailed, constructive feedback. We have carefully examined the concerns you raised and are committed to revising the paper as thoroughly as possible to address them. Below, we respond to your main concerns regarding **(1) the limited range of datasets** and **(2) the restricted set of baseline models**.
>
> ### Addressing concerns about the evaluation scope (Data, Baselines)
>
> To address the concern that restricting the main experiments to DR datasets may weaken the **generality claim** of the framework, and that **a broader set of baseline (comparison) models** is needed, we made the following revisions.
>
> First, we conducted additional **small-scale domain-incremental experiments** using three public skin cancer datasets and report the results in Table 6. We also expanded the baselines to include **prompt-based CL**, **regularization-based CL**, **rehearsal-based CL**, and **architecture-based CL** methods, and updated Table 1, Table 4, and Table 6 accordingly to cover a broader range of CL approaches. The dataset splits and implementation details for these new experiments are described **in detail in Appendices G and H** to further improve reproducibility.
>
> As shown in Table 1 and Table 4, even after expanding the set of baselines, the proposed method still achieves the best performance. In the skin cancer domain-incremental setting, it also achieves the **highest AvgACC (0.732)**, demonstrating that the framework can be **extended beyond DR to other diseases and imaging domains**. Furthermore, by introducing the CARA (Cost-Adjusted Retained Accuracy) metric that jointly considers computational cost, we quantitatively show that our method offers higher practical utility (accuracy per compute) under comparable budget. Specifically, the proposed method achieves \( \mathrm{CARA}_{0.5} = 0.105 \), surpassing strong baselines such as OS (0.101), OS++ (0.095), DER++ (0.050), Coda-Prompt (0.059), and MoE-Adapters (0.056), thereby supporting our claim that it **retains and delivers more accuracy under the same or similar compute budget**.
>
> Here, **CARA** combines AvgACC with a GFLOPs-based penalty term and **summarizes the trade-off between accuracy and compute into a single scalar** metric. The **exact formulation, computation procedure, and interpretation** of CARA are described in Appendix B.
>
> Finally, we have carefully reviewed the references you suggested and incorporated them at appropriate places in the main text to strengthen the related discussion. Thank you again for your insightful feedback.

---

### Author Response · Authors · 2025-11-19
**Overall Response and Summary of Changes**

First, we would like to express our sincere gratitude to the reviewers **uosG**, **zHwv**, **XbYJ**, and **Ab34** for their careful evaluation and constructive feedback. In order to faithfully address the concerns raised, this response first **summarizes the strengths and concerns highlighted in each review**, and then **organizes and presents the modifications and improvements we have made in response to these concerns**.

## Summary of Strengths

1. **Clear motivation for Continual Learning (CL) in the medical domain**
   Under the realistic constraint of restricted data sharing, the paper clearly identifies limitations of existing PCL methods with concrete examples, and proposes a new training strategy designed to address these issues.

2. **Simultaneous improvement in efficiency and performance**
   Compared to the SOTA method (e.g., OS-Prompt++) with **dual inference**, our method achieves **approximately 33% reduction in GFLOPs** while still yielding **+10 points in Accuracy**, **+9 points in F1-score**, and better performance on key CL metrics (e.g., BWT/AvgF).

3. **Well-motivated problem and clear supporting rationale**
   The manuscript presents a clear research motivation and supporting rationale, and multiple reviewers commented positively on the overall clarity and completeness of the writing and exposition.

## Summary of Concerns

1. **Limited experimental scope** (three DR classification datasets, short sequence length)
   **Response:** To ensure that the framework is not restricted to DR, we conducted additional small-scale pilot experiments using three public skin cancer datasets in a domain-incremental setting, and report that the proposed method extends well to other medical domains. The corresponding results are presented in Table 6, and the dataset details and experimental setup are described in Appendix H.

2. **Insufficient baselines** (missing comparisons with other CL families)
   **Response:** Based on the references and prior works suggested by the reviewers, we additionally include comparisons with MoE-Adapters, DER++, and Online EWC. Following the structure in the Related Work section, we expand the baselines to include **prompt-based CL**, **regularization-based CL (EWC/SI)**, **rehearsal-based CL**, and **architecture-based CL** methods. We then report performance, forgetting metrics, and efficiency under the same setting in Table 1 and Table 4, showing that our method consistently outperforms a wide range of baselines.

3. **Inconsistent reporting of key continual-learning metrics** (BWT/AvgF inconsistencies between main text and appendix)
   **Response:** In the main table (Table 4), we now **report BWT, AvgF, and GFLOPs for all baselines**, and additionally introduce the CARA metric to summarize the relationship between performance and GFLOPs. A detailed description of CARA is provided in Appendix B.

4. **Reproducibility of data processing and training procedure** (e.g., APTOS 5 -> 3 class mapping)
   **Response:** To strengthen reproducibility of data preprocessing and training, we added *UniPrompt-CL Training Procedure* in Appendix F, and provide detailed hyperparameter configurations for all baselines in Appendix G. We also explicitly describe the APTOS **\5 -> 3 class mapping table** and preprocessing pipeline in the data processing section of Appendix A. Furthermore, we provide a link to an **anonymous code repository/scripts** (including model configuration, training schedules, and data splits) to ensure **full reproducibility**, and plan to additionally include the extra small-scale data partitions, whose detailed dataset splits are summarized in Appendix H. Since updating an existing anonymous repository may be constrained by ICLR policy, we will update it at the camera-ready stage if required by the AC and policy.

We identify the above four points as the primary concerns and **provide concrete improvements for each**. All new and revised content has been incorporated into appropriate locations in the main paper. If there are any additional issues or concerns that we may have missed, we would be happy to address them promptly and thoroughly.

---

> ### Author Response · Authors · 2025-11-19
> **Summary of Key Updated Tables Referenced in the Rebuttal**
>
> To more clearly address the reviewers’ concerns regarding evaluation scope, baseline coverage, and the efficiency–performance trade-off, we reproduce below the three key updated tables (Tables 1, 4, and 6) that are referenced in our responses. The final versions of these tables can be found in the revised manuscript.
>
> **Table 1.** Final Accuracy (Acc) and F1-Score (F1) results after the final stage and performance comparison with other PCL models.
> The symbol $^\dagger$ denotes our proposed model. PCL denotes prompt-based continual learning; Arch-CL refers to architecture-based continual learning; Reh-CL means rehearsal-based continual learning; and Reg-CL indicates regularization-based continual learning. (**Bold** indicates the best performance; scores are mean values with negligible deviations.)
>
> | **CL-Type (Ref)**        | **Model**              | **APTOS Acc** | **APTOS F1** | **DDR Acc** | **DDR F1** | **DRD Acc** | **DRD F1** |
> |--------------------------|------------------------|---------------|--------------|-------------|------------|-------------|------------|
> | PCL (ECCV2024)           | OS                     | 0.687         | 0.637        | 0.693       | 0.648      | 0.619       | 0.568      |
> | PCL (ECCV2024)           | OS++                   | 0.743         | 0.686        | 0.697       | 0.655      | 0.623       | 0.565      |
> | Arch-CL (CVPR2024)       | MoE-Adapters           | 0.835         | 0.742        | 0.747       | 0.694      | 0.564       | 0.478      |
> | PCL (CVPR2023)           | Coda-Prompt            | 0.682         | 0.646        | 0.721       | 0.697      | 0.663       | 0.557      |
> | PCL (CVPR2022)           | L2P                    | 0.353         | 0.174        | 0.421       | 0.194      | 0.603       | 0.252      |
> | PCL (ECCV2022)           | Dual-prompt            | 0.363         | 0.185        | 0.435       | 0.222      | 0.604       | 0.259      |
> | Reh-CL (NeurIPS2020)     | DER++                  | 0.531         | 0.442        | 0.609       | 0.567      | 0.681       | 0.612      |
> | Reg-CL (ICML2018)        | Online EWC             | 0.746         | 0.695        | 0.702       | 0.708      | 0.698       | 0.653      |
> | -                        | **UniPrompt-CL$^\dagger$** | **0.849**     | **0.761**    | **0.772**   | **0.723**  | **0.701**   | **0.656**  |
>
> ---
>
> **Table 4.** Performance evaluation of AvgACC, BWT, and Cost-Adjusted Retained Accuracy (CARA) across three diabetic retinopathy datasets. Training-time GFLOPs per step (including forward and backward passes). The symbol $^\dagger$ denotes our proposed model. The best results are highlighted in **bold**.
>
> | **Method**               | AvgACC$\uparrow$ | BWT$\uparrow$ | AvgF$\downarrow$ | GFLOPs$\downarrow$ | $\mathrm{CARA}_{0.5}\uparrow$ |
> |--------------------------|------------------|---------------|------------------|---------------------|-------------------------------|
> | OS-Prompt                | 0.666            | -0.132        | 0.132            | **34.26**           | 0.098                         |
> | MoE-Adapters             | 0.716            | -0.080        | 0.080            | 105.64              | 0.064                         |
> | Coda-Prompt              | 0.688            | -0.140        | 0.140            | 134.33              | 0.051                         |
> | L2P                      | 0.459            | -0.296        | 0.296            | 116.93              | 0.029                         |
> | Dual-prompt              | 0.467            | -0.291        | 0.291            | 105.05              | 0.032                         |
> | DER++                    | 0.607            | -0.288        | 0.288            | 168.02              | 0.033                         |
> | Online EWC               | 0.715            | -0.174        | 0.174            | 100.62              | 0.059                         |
> | OS-Prompt++ (Original)   | 0.769            | -0.113        | 0.113            | 51.12               | 0.095                         |
> | OS-Prompt++ (Add Dino-v2)| 0.812            | -0.125        | 0.125            | 66.42               | 0.087                         |
> | **UniPrompt-CL$^\dagger$** | **0.844**      | **-0.079**    | **0.079**        | 44.17               | **0.116**                     |

---

> > ### Author Response · Authors · 2025-11-19
> >
> > ---
> >
> > **Table 6.** Performance evaluation of AvgACC, BWT, AvgF, GFLOPs, and CARA across three small external skin cancer datasets. The symbol $^\dagger$ denotes our proposed model.
> >
> > | **Model**        | **AvgACC$\uparrow$** | **BWT$\uparrow$** | **AvgF$\downarrow$** | **GFLOPs$\downarrow$** | $\mathrm{CARA}_{0.5}\uparrow$ |
> > |------------------|----------------------|-------------------|----------------------|-------------------------|-------------------------------|
> > | OS               | 0.682                | -0.135            | 0.135                | **34.26**               | 0.101                         |
> > | OS++             | 0.725                | -0.063            | 0.063                | 51.12                   | 0.095                         |
> > | MoE-Adapters     | 0.597                | -0.040            | 0.040                | 105.64                  | 0.056                         |
> > | Coda-Prompt      | 0.713                | -0.041            | 0.041                | 134.33                  | 0.059                         |
> > | Dual-prompt      | 0.637                | **-0.012**        | **0.012**            | 105.05                  | 0.061                         |
> > | DER++            | 0.722                | -0.099            | 0.099                | 168.02                  | 0.050                         |
> > | Online EWC       | 0.708                | -0.157            | 0.157                | 100.62                  | 0.060                         |
> > | **UniPrompt-CL$^\dagger$** | **0.732**    | -0.049            | 0.049                | 44.17                   | **0.105**                     |

---

### Author Response · Authors · 2025-12-01
**AC Meta-Summary: Closing Major Concerns with Concrete Updates**

**Dear Area Chair**, thank you for overseeing our submission and the discussion.

In the rebuttal, **we summarized the strengths and key concerns regarding our paper above, and focused on fully addressing those concerns through concrete updates in the revised manuscript (not just arguments).**

**1.What we changed (with explicit pointers)**

•	**Scope / generality**: beyond the 3 DR datasets, we added a domain-incremental pilot on three public skin-cancer datasets to test transfer beyond fundus classification (Table 6, Appendix H).\
•	**Baseline completeness**: we expanded comparisons to cover prompt-based, rehearsal-, regularization-, and architecture-based CL (incl. DER++, Online EWC, MoE-Adapters) under consistent settings (Tables 1, 4, 6; Appendix G).\
•	**CL metrics consistency**: we resolved the BWT/AvgF inconsistency by reporting AvgACC, BWT, AvgF, and GFLOPs for all baselines in the main evaluation, and introduced CARA to summarize the efficiency–retention trade-off (Table 4;Table 6; Appendix B).\
•	**Reproducibility / clarity**: we added stage-wise pseudocode for the full training procedure (prompt freezing + fixed-ratio prompt expansion), detailed preprocessing (incl. APTOS 5→3 mapping), and baseline hyperparameters (Appendices A, F, G), alongside an anonymous code/scripts link.

**2.Why we believe the contribution is not merely incremental (core technical)**

UniPrompt-CL removes the typical reliance on an extra query module / dual inference by:
1.	using a single layer-shared unified prompt pool enabling cross-layer reuse,\
2.	enforcing fixed per-stage prompt growth to keep compute predictable, and\
3.	adding a regularizer that increases utilization/learning of newly added prompts, validated via ablations (e.g., Table 5) and reflected in consistent gains in accuracy, forgetting metrics, and CARA (e.g., Table 4).

**3. Framing tightened (scope clarity)**

Finally, we tightened the framing: rather than claiming universal “medical AI” coverage, we explicitly scope the method to domains with standardized acquisition / constrained viewpoints and fine-grained discriminative cues, supported by the added cross-domain pilot and qualitative analyses (Appendix C; new Fig. 4). Notably, after these additions, one reviewer explicitly stated that the expanded datasets and CL-family comparisons addressed their concerns and raised their rating.

**We respectfully hope the revised manuscript demonstrates a clear, evidence-backed method contribution and resolves the primary reasons for doubt. Thank you again for your time and consideration.**

---

### Meta-Review · Area_Chair_8yiz · 2025-12-19

**Summary:**

Reviewers agreed that the proposed prompt-based continual learning method is reasonably motivated for the medical domain at hand. They agree that there are empirically observable gains and computational advantages underlying the proposition. They generally have a mixed assessment on whether the method itself is described well, and there is a more general concern on the incremental and applied nature of the work in extension to prior art. This limitation in scope has also led to multiple reviewers' concerns over limited experimentation, inadequate comparisons, and overly broad claims regarding finding solutions to the medical field and "sustainability".
As described below, the AC agrees with many of these aspects and does not find the latter concerns to have been resolved in the rebuttal.

**Reviewer Concerns:**

Reviewers concerns spanned from limited empirical analysis, to overly broad conclusions, to incremental technical contributions beyond domain application. Going through the rebuttal and general revision comments, the AC can see that a great attempt has been made to include additional results and further comparisons. There are good steps towards generally improving the paper.
The part that is less clearly resolved, and where concerns largely remain open, is the scope and technical contribution of the work. Some reviewers point out that this paper would perhaps be better suited to an applied medical venue, but at least three reviewers clearly point out that there is a lack of differentiation from various prior works. This results in consensus that the method is at the worst an incremental application of largely existing methods, or at the best an improvement that isn't sufficiently differentiated in the way it's described.
Reading through the original paper, the differentiation of the proposed method in terms of technical advancement is also not immediately clear to the AC. The AC has seen the general comment by the authors in this regard, but agrees with the reviewers that this point would require much more in-depth substantiation to clarify the valid concerns.

**Reviewer Scores:**

Reviewer scores were mixed in that two reviewers gave borderline scores, and two reviewers gave clear reject recommendations. Two of the reviewers were able to take part in the discussion prior to the freeze: one of which chose to retain their reject rating and one of which would have chosen to raise their initial score of 4 (which is not displayed but acknowledged by the AC). As such, the scores at that stage would still be mixed, with likely two times 6 and two ratings of two. As argued above, the AC believes it unlikely that any further discussion, and also discussion with the other two reviewers, would result in any substantial change towards unanimously recommending acceptance or one reviewer strongly championing the paper. Whereas some concerns could be addressed with some additional empirical results, the overall questions regarding the main contributions and the technical advancement are unlikely to be resolved through discussion alone.

---

### Decision · Program_Chairs · 2026-01-26

Reject